# Morphological, Morphometrical and Radiological Features of the Pelvic Limb Skeleton in African Green Monkeys (*Chlorocebus sabaeus*) from Saint Kitts and Nevis Islands

**DOI:** 10.3390/ani15020209

**Published:** 2025-01-14

**Authors:** Cristian Olimpiu Martonos, Alexandru Ion Gudea, Gilda Rawlins, Florin Gheorghe Stan, Calin Lațiu, Cristian Constantin Dezdrobitu

**Affiliations:** 1Department of Biomedical Sciences, Ross University School of Veterinary Medicine, Basseterre P.O. Box 334, Saint Kitts and Nevis; cmartonos@rossvet.edu.kn (C.O.M.); grawlins@rossvet.edu.kn (G.R.); cdezdrobitu@rossvet.edu.kn (C.C.D.); 2Department of Anatomy, Faculty of Veterinary Medicine, University of Agricultural Sciences and Veterinary Medicine Cluj-Napoca, Romania, Calea Mănăştur 3-5, 400372 Cluj-Napoca, Romania; florin.stan@usamvcluj.ro; 3Faculty of Animal Husbandry and Biotechnologies, University of Agricultural Sciences and Veterinary Medicine Cluj-Napoca, Romania, Calea Mănăştur 3-5, 400372 Cluj-Napoca, Romania; calin.latiu@usamvcluj.ro

**Keywords:** vervet monkey, osteology, radiography, gross anatomy, osteometry, radiology

## Abstract

The present paper summarizes the detailed gross morphological and imagistic features of the pelvic skeletal elements alongside some radiographical interpretations. The added metrical data makes the anatomical description of the pelvic skeleton in *Chlorocebus sabaeus* more accurate and precise concerning these monkeys’ anatomy. Several morphological features of each of the studied skeletal elements explain the different adaptative changes in this species as well, as it shows, in a comparative way, different sets of elements in the primate skeletal anatomy.

## 1. Introduction

The anatomical, physiological, and physio-pathological similarities observed between the non-human primates and humans made these species highly appreciated in the biomedical area [1]. Numerous anatomical studies involved the pelvic limb bones of non-human primates [2,3,4,5,6,7,8,9,10] and humans [11,12,13] in an attempt to find correlations between these structures’ morphological aspects and these species’ locomotor behavior.

Even if for most quadrupedal mammals for their locomotor behavior, the thoracic limbs are more important, as they support most of their overall body weight compared to the pelvic limb [14,15,16,17,18]; in primates, this feature is reversed. According to the reported data, these species support their body weight on the pelvic limb and this is seen as an adaptation to arboreal locomotion [16,19,20,21].

The African green monkeys (*Chlorocebus sabaeus*) or vervet monkeys (AGM’s) are opportunistic primates living in Sub-Saharan Africa [22,23]. In the 17th century, these monkeys were transported to Saint Kitts and Nevis on the first ships of slaves [24,25]. The absence of natural predators and the lack of common pathogens, corroborated by the high adaptability of these monkeys, made them exceed the number of humans on the island and become a real problem, “the monkey problem” for the local authorities [15,23,25]. However, even though the vervet monkeys are a real pest for agriculture, the presence of these monkeys on the island contributes to the local economy being used by the local people for tourism. Also, on the island, there are two accredited biomedical facilities that use the African green monkeys for preclinical or clinical research projects [25,26].

Genetically, the vervet monkey population from Saint Kitts is linked to the *C. sabaeus* population from Gambia, a region in West Africa. Also, genomics studies, which compared the DNA of AGM’s with the human genome, reported a similarity of 90% between these two species [27,28,29,30,31]. Due to these genetic similarities with humans, the *C. sabaeus* monkeys have already been used as animal models in cardiac transplants and cardiac disease and in research projects on pathogens such as *Mycobacterium tuberculosis*, *Staphylococcus aureus*, *Yersinia pestis*, *Klebsiella pneumoniae* and several viruses [23,32,33].

Their destructive behavior related to the local farmers’ production, the interactions with the tourists, and the maintenance in captivity for research purposes can be linked to the occurrence of several bone pathologies or even traumatic fractures. Although these monkeys are used as animal models, the anatomical knowledge related to them is quite incomplete, especially in terms of their skeletal anatomy [23,34]. These facts have led us to think that a gross anatomical description of the pelvic limb skeleton is necessary, making it this paper’s main purpose, as the literature lacks these basic descriptive anatomic features.

## 2. Materials and Methods

The studied specimens were part of a private collection composed of five skeletons. Three of them were males (K920, K 930, K945), and two were females (A585 and V438), in different states of preservation. One fresh adult male cadaver (donated by the Pathology Department) was used to obtain radiographs. The study was developed in the anatomy laboratory of Ross University School of Veterinary Medicine, Saint Kitts and Nevis, Basseterre, compliant with IACUC regulations (TSU10.27.2023 CM) from the RUSVM Institutional Animal Care and Use Committee. All six specimens were declared adult specimens.

To obtain relevant information, each bone of the corresponding pelvic limbs was photographed in a standard position with a measuring scale placed nearby.

Image capturing was performed with a DSLR, Canon EOS 90D (Canon, Melville, NY, USA). Small improvements (black background, scale bar) were made to the photos using Adobe Photoshop^®^21 (Adobe Itl, Atlanta, GA, USA).

Each bone was carefully assessed and described. All anatomical structures were named using the Nomina Anatomica Veterinaria 2017 terminology [35]. Radiographs were used for position confirmation and structure identification. In the Ross University School of Veterinary Medicine veterinary clinic, radiographs were performed using Vet Ray Technology Standard Vet device (Niles, IL, USA), with 85 kVp, 3.2 mAs radiographic settings.

The entire activity respected all the radiological protection rules.

The different areas of the pelvic limb were radiographed in standard positions on the radiograph table. For the pelvic area, ventrodorsal, dorsoventral, and laterolateral projections were performed. For the thigh and crus areas, latero-medial, medio-lateral, cranio-caudal, and caudo-cranial projections were performed and for the pes area, latero-medial, medio-lateral, dorso-plantar, and planto-dorsal projection were performed. The bony structures were observed and identified on radiographic images and compared with the gross anatomical images.

Osteometry is an important part of the evaluation of the general anatomy in all human and animal species. The different metrical data may play a key role in evaluating the stature and conformation of individuals and their forensic importance. Morphometric measurements of the bones were initially adapted after the general set of measurements used in animal osteometry with a series of specific measurements used in anthropology and human osteometric investigations [36,37,38,39,40,41,42] (Table 1, Table 2, Table 3 and Table 4). Each measurement is briefly described in the tables and a codification is given. For the statistical evaluation, some basic functionalities of Google Sheets were used (average, standard deviation).

## 3. Results

After the examination of the skeletons, we can report that in African green monkeys (AGMs or vervet monkeys), the pelvic limb skeleton is built by the coxal bones, femur, patella, tibia, fibula, tarsal, metatarsal, phalanges, and sesamoid bones.

The fusion between right and left *Ossa coxae* (Figure 1, Figure 2, Figure 3 and Figure 4, Table 5) forms the pelvic girdle. The anatomical features of the coxal bones provide the opportunity to describe the presence of three bones: ilium *(Os ilium*), ischium (*Os ischii*), and pubis (*Os pubis*). Because of their anatomy-topographical disposition, at the junction point of all these bones a deep cavity is formed—the acetabulum (*Acetabulum*) (Figure 1, Figure 2 and Figure 4). This cavity receives the head of the femur and is part of the hip joint (*Articulatio coxae*).

Even if the margin of the acetabulum (*Margo acetabularis*) (Figure 1 and Figure 2) is well developed in cranial, dorsal, and caudal segments, it is incomplete in its ventro-medial area and forms the acetabular notch (*Incisura acetabuli*) (Figure 1 and Figure 2). This notch continues medially with the acetabular fossa (*Fossa acetabuli*) and is surrounded by an evident *facies lunata* (Figure 1 and Figure 2).

Dorsally from the acetabulum, a very low ischiatic spine (*Spina ischiadica*) (Figure 1, Figure 2 and Figure 3) can be observed. The greater ischiatic notch (*Incisura ischiadica major*) and the lesser ischiatic notch (*Incisura ischiadica minor*) are not very pronounced. The groove of the internal obturator muscle’s tendon was visible for all the specimens. Also, the caudo-medial margins of the pubis, alongside the cranial margin of the ischium, delimit the elliptically shaped obturator foramen (*Foramen obturatum*) (Figure 1, Figure 3 and Figure 4). The cranial end of this foramen is marked by a delicate obturator sulcus, which accommodates the obturator nerve (*N. obturatorius*), in its trajectory toward the adductor muscles.

In AGM (vervet monkey), the wing of the ilium (*Ala ossis ilii*) (Figure 1, Figure 2, Figure 3 and Figure 4) has a rectangular shape and is longer than the body of the bone (*Corpus ossis illi*). The gluteal surface (*Facies glutea*) is concave and is bordered cranially by a roughened and cranially convex iliac crest (*Crista iliaca*). Ventro-caudally, the iliac crest continues with a straight cranial ventral iliac spine (*Spina iliaca ventralis cranialis*).The junction point of these two borders—the tuber coxae (*Tuber coxae*)—is sharp and discrete. Caudally, the ventral border of the ilium (*Spina iliaca ventralis caudalis*) allows the identification of a well-developed lateral area of the rectus femoris muscle (*Area lateralis m. recti femuris*), located cranially to the acetabulum. The *labium internum* and *labium externum* of the ventral iliac spine were visible in all studied specimens.

The tuber sacrale (*Tuber sacrale*) (Figure 1, Figure 2 and Figure 3) was rounded and not very well-developed. The caudal segment of the dorsal border of the ilium (*Spina iliaca dorsalis caudalis*) has a concave aspect and forms the greater ischiatic notch.

The medial surface (*Facies sacropelvina*) (Figure 2) was convex and articulated with the wings of the sacrum. The auricular surface (*Facies auricularis*) was well-developed and continues cranio-dorsally with the iliac tuberosity (*Tuberositas iliaca*) and cranially with the iliac fossa (*Fossa iliaca*). Ventro-caudal, an evident arcuate line (*Linea arcuata*) (Figure 1) with a discrete tubercle of the psoas minor muscle (*Tuberculum m. psoas minoris*) could be observed. Caudally, the body of the ilium ends at the level of the acetabulum.

The ischium (Figure 1, Figure 2, Figure 3 and Figure 4) is a rectangular flat bone which forms the ventro-caudal part of the coxae bone. In a sagittal plane, the ramus of the right and left ischial bones (*Ramus ossis ischii*) articulate together and form the ischial symphysis (*Symphysis ischiadica*) (Figure 1), which is the caudal segment of the pelvic symphysis. The caudo-lateral extremity shows a prominent structure, the ischial tuberosity (*Tuber ischiadicum*) (Figure 1). The cranial border of the body of the ischium (*Corpus osiis ischii*) forms the caudo-lateral margin of the obturator foramen. The dorso-lateral border shows the lesser ischiatic notch, and a groove for the tendon of the internal obturator muscle can be observed immediately caudal by the ischiatic spine. The ischial table (*Tabula ossis ischii*) had a concave aspect on its dorsal surface. Caudo-medial borders have a ventromedial direction and form together the ischiatic arch (*Arcus ischiadicus*), which has a V-shaped aspect.

The pubis (Figure 1, Figure 2, Figure 3 and Figure 4) is a V-shaped bone and comprises the ventral and caudal segments of the coxae bone. The cranial ramus (*Ramus cranialis ossis pubis*) runs in a mediolateral direction and fuses with the caudal end of the body of the ilium and the cranial segment of the body of the ischium at the level of the acetabulum. The cranial ramus of the pubis, near its junction with the ilium, shows a small ilio-pubic eminence (*Eminencia iliopubica*) (Figure 1). Medially to the ilio-pubic eminence, the *pecten ossis pubis* (Figure 1) was observed. The caudal ramus of the pubis (*Ramus caudalis ossis pubis*) runs in a cranio-caudal direction and medially fuses with the opposite bone and forms the pubic symphysis (*Symphyses pubica*), the cranial segment of the pelvic symphyses (*Symphyses pelvina*) (Figure 1). The junction point of these two rami of pubic bone is evident and forms the body of the pubis (*Corpus ossis pubis*). The pubic tubercle (*Tuberculum pubicum*) (Figure 1) is well developed and represents the most cranial point of the pelvic symphyses (Table 5).

Radiographically, all the anatomical components of the coxae bones (Figure 3 and Figure 4) could be identified. Many morphological anatomical features of the ilium, ischium, and pubis bone were visible in the ventrodorsal, dorsoventral, and laterolateral projections. Even so, some anatomical structures on the skeleton were easily noticeable, while on radiographs, due to the superposition, were difficult to analyze. In this situation, the ischiatic spine, the elements of the acetabular cavity or the articular areas, located on the medial surfaces of the wings of the ilium, were hardly visible.

Available measurements are listed in Table 5. Some specimens were heavily degraded, which prevented some of the standard measurements from being taken. Although the number of available measurements is minimal, the mean values for males and females do not show significant differences in terms of the overall length of the pelvis and length of the acetabulum (P1, P2/3), maintaining a 10–20% difference in favor of male dimensions for most of the taken measurements (Table 5), but indicating some interesting hints on the data related to the size of the symphysis (P4) and height of the iliac body (P5), which showed higher values for females, similarly to the ones confirming the subjective observation on the shape of the obturator foramen (P8/P8a—indicating a more oblong overall shape for females than in males). Another value that is higher for female examined specimens is the area of the auricular surface (P22).

The femur (Figure 5) is the anatomic base of the thigh region and is the longest bone of the higher part of the limb in the African green monkey. It shows well-developed proximal and distal ends and a long cylindrical shaft (*Corpus femuris*) (Table 6).

In all studied specimens, a small cranial convexity of the shaft could be observed. The femoral head (*Caput ossis femuris*) (Figure 5) is the articular structure of the proximal end and has a spherical aspect with a medial orientation. This structure articulates with the acetabulum. The *Fovea capitis* has an elliptical aspect with a transversal diameter smaller than the longitudinal diameter and a ventrocaudal position regarding the center point of the femoral head. Between this structure and the acetabular fossa, the ligament of the head of the femur (*Lig. capitis ossis femoris*) can be observed in fresh specimens. The neck of the femoral head (*Collum ossis femoris*) (Figure 5) is short and cylindrical (Table 6). On the lateral aspect of the proximal end of the femur a very well-developed greater trochanter (*Trochanter major*) (Figure 5) is visible, pointing higher than the level of the femoral head. In a caudo-medial position, immediately distally to the femoral head, the lesser trochanter (*Trochanter minoris*) can be identified. On the caudal surface of the proximal epiphysis, between the greater and lesser trochanter, a well-developed intertrochanteric crest (*Crista intertrochanterica*) (Figure 5) can be observed. On the inner surface of this crest, a deep trochanteric fossa (*Fossa trochanterica*) (Figure 5) is noticeable in all studied specimens. On the cranial side of the proximal epiphysis, a thin intertrochanteric line (*Linea intertrochanterica*) (Figure 5) is visible in the studied specimens. The third trochanter (*Trochanter tertius*) could not be noticed. Most of the specimens were measurable, with data listed in the following table (Table 6).

On the lateral surface of the femoral diaphysis, the distal end of the intertrochanteric crest shows a small bony eminence, the gluteal tuberosity (*Tuberositas glutea*) (Figure 5). In some AGM (vervet monkey) specimens, this structure continues distally, on the caudo-lateral part of the shaft of the femur, with a very clear *Linea aspera* (Figure 5). Between the proximal end of the *Linea aspera* and the base of the lesser trochanter an oblique bony crest could be observed, the *Linea pectinea* (Figure 5).

The articular structures of the distal end of the femur have a double disposition. The cranial aspect is covered by the femoral trochlea (*Trochlea ossis femoris*) (Figure 5 and Figure 6), and the distal–caudal aspect is covered by the medial and lateral femoral condyles (*Condylus medialis* and *Condylus lateralis*) (Figure 5 and Figure 6).

The medial condyle, in all specimens, was larger and more well-developed than the lateral condyle (Table 6) and both allowed the identification of their epicondyles (*Epicondylus medialis* and *Epicondylus lateralis*) (Figure 5). A deep popliteal fossa *(Fossa m. poplitei*) can be observed on the abaxial surface of the lateral epicondyle. A very discrete ligamentary fossa (Figure 5) can be observed in the same area between the lateral femoral condyle and the lateral trochlear ridge. An intercondylar fossa (*Fossa intercondylaris*) (Figure 5) is present in the axial plane, between the medial and lateral femoral condyles. Dorsally to this structure, on the distal end of the femoral shaft, a triangular smooth area allowed the identification of the popliteal surface (*Facies poplitea*) (Figure 5). The medial supracondylar line (*Linea supracondylaris medialis*) (Figure 5), lateral supracondylar line (*Linea supracondylaris lateralis*) (Figure 5), and the intercondylar line (*Linea intercondylaris*) (Figure 5) surround the popliteal surface. The caudo-proximal aspect of the medial and lateral femoral condyles bears small circular areas, *Facies articularis sesamoidea medialis* and *Facies articularis sesamoidea lateralis*, for articulation with the sesamoid bones of gastrocnemius (*Ossa sesamoidea m. gastrocnemii*) (Figure 5). A reduced asymmetry can be noticed at the level of the femoral trochlea, the lateral lip being larger than the medial lip (Table 6).

In African green monkeys, the patella (*Patella)* (Figure 7) is a relatively pentagonally shaped bone. The articular surface (*Facies articularis*) (Figure 7) comes in contact with the trochlea of the femur. Its cranial surface (*Facies cranialis*) (Figure 7) is covered by skin. The apex (*Apex patellae*) (Figure 7) is triangular and has a ventral orientation. The base (*Basis patellae*) (Figure 7) forms the proximal end, connected with the apex of the patella by a medial, respectively, lateral patellar margin. Usually, the lateral margin is straight, but the medial margin shows a medial bony projection.

The radiographs of the thigh area of the cranio-caudal and caudo-lateral projections were very helpful. These images provided information related to the position, presence, and size of the anatomical features of the proximal and distal epiphysis. Structures such as the greater trochanter, lesser trochanter, intertrochanteric crest, and subtrochanteric fossa could be easily observed at the level of the proximal epiphysis of the femur (Figure 3 and Figure 4). The more delicate structures of this extremity, such as the gluteal tuberosity, *linea aspera* and *linea pectinea*, were not identifiable. For the distal epiphysis, the radiographic imaging confirmed the presence of the sesamoid bones of gastrocnemius muscles—two round bony structures located caudal and proximal from thefemoral condyles (Figure 6).

The metrical data collected for the femur (Table 6) comprises sets that are a little more complete than the ones in the pelvic bone. The values that stand out (with the reserve of the reduced amount of available data) are the ones for the head angulation (F17) as well as the ones for the width of the medial condyle (F19), indicating very similar sets of data, opposite to most of the other metric data that all pinpoint to the expected sex-dependant variation seen in the pelvic bone (slightly smaller sets of values with 10–15% for females). More than that, the common indices calculated for the entire length of the bone (proximal index, diaphyseal index, and distal index) show very little differentiation among males and females.

The anatomical base for the crural area (*Skeleton cruris*) is formed by two long bones, the tibia (*Tibia*) and fibula (*Fibula*) (Figure 8). Anatomo-topographically, the tibia has a medial placement. It is more developed than the fibula (which is less developed and has a lateral disposition). Between these bones, an ample interosseous space can be seen.

The proximal end of the tibia is wider and has a triangular aspect, while the distal end is almost circular. The articular structures of the proximal end are the medial and lateral condyles of the tibia (*Condylus medialis* and *Condylus lateralis*) (Figure 8), which articulate with the medial and lateral femoral condyles. The articular surfaces (*Facies articularis proximalis*) (Figure 8, Table 7) of the tibial condyles bear two small excavations. More than that, a visible asymmetry could be observed, the medial articular surface being slightly larger with a pronounced concavity, compared with the lateral articular surface. Between them, in a cranio-caudal direction, the intercondylar area (*Area intercondylaris*) (Figure 8) is well-developed. At these levels, three segments are visible: the cranial intercondylar area (*Area intercondylaris cranialis*), the origin points for the cranial cruciate ligament (*Lig. cruciatum craniale*), the central intercondylar area (*Area intercondylaris centralis*), and the caudal intercondylar area (*Area intercondylaris caudalis*), which is the origin point for the caudal cruciate ligament (*Lig. cruciatum craniale*).

This last segment continues caudally with the popliteal notch (*Incisura poplitea*) (Figure 8). Into the central intercondylar area, the two intercondylar tubercles (*Tuberculum intercondylare mediale* and *Tuberculum intercondylare laterale*) (Figure 8) are well-developed and form a prominent intercondylar eminence (*Eminentia intercondylaris*) (Figure 8). In all specimens, the lateral intercondylar tubercle was slightly taller than the medial one. The well-developed tibial tuberosity (*Tuberositas tibiae*) (Figure 8) is the cranial continuation of the cranial intercondylar area. Distally, it continues with the cranial margin (*Margo cranialis*) (Figure 8).

The caudo-lateral surface of the lateral tibial condyle provides a circular articular surface for the head of the fibula (*Facies articularis fibularis*) (Figure 8). The cranio-lateral margin owns a very discrete extensor’s groove (*Sulcus extensorius*) (Figure 8).

The body of the tibia (*Corpus tibiae*) (Figure 8) has a triangular aspect in its proximal half and is circular in its distal half. All three surfaces—the medial, lateral, and caudal were visible (*Facies medialis*, *Facies caudalis*, *Facies lateralis*). Between them, we could identify three margins (*Margo cranialis*, *Margo medialis*, and *Margo lateralis*). The caudal surface is crossed by a thin oblique bony line which is the insertion point for the popliteus muscle (*Linea m. poplitei*) (Figure 8). The proximal end of the lateral surface has an excavated area, which accommodates the muscular bellies of the flexor muscles of the tarsus and extensors of the digits.

The distal end of the tibia bears the articular elements which are part of the talocrural joint (*Articulatio tarsocruralis*) and part of the distal tibiofibular joint (*Articulatio tibiofibularis distalis*).

The central segment of the distal end shows two grooves and a median crest that form the tibial cochlea (*Cochlea tibiae*) (Figure 8), which articulates with the trochlea of the talus, being part of the most mobile joint of the tarsal area. The medial malleolus (*Malleolus medialis*) (Figure 8) is visible, and on its caudal surface, a deep sulcus malleolaris could be identified. The distal end of the lateral surface of the tibia was incomplete and a fibular notch (*Incisura fibularis*) (Figure 8) was noted in this area.

The measurements of the tibia and fibula are listed below (Table 7). Due to heavy damage on some of the bone parts, some measurements were impossible.

The metrical data collected for tibia show, in almost all cases, the expected metrical differenced among males and females. In all sets, there is a small amount of differentiation, with data for females representing values of 80–97% of the values recorded for males. The value recorded for the area of the tibial plateau (T7) is the only one that shows a higher difference among males and females, with recorded values for females representing 70% of the data recorded for males. This value should be interpreted with much caution, with the data set being largely reduced for female specimens (n = 3).

In African green monkeys, the fibula (*Fibula*) (Figure 8, Table 7) is a long bone with a cylindrical aspect in a transversal section. The head of the fibula (*Caput fibulae*) (Figure 8), has a tuberous aspect and bears an articular surface (*Facies articularis capitis fibulae*) (Figure 8) for articulation with the lateral condyle of the tibia. This is the proximal tibiofibular joint (*Articulatio tibiofibularis proximalis*). The neck of the fibula (*Collum fibulae*) was the distal continuation of the head and was a narrow segment located between the fibular head and the fibular body (*Corpus fibulae*) (Figure 8). All four surfaces of the fibular body (*Facies medialis*, *Facies lateralis*, *Facies caudalis*, *Facies cranialis*) were clear and easy to identify, separated by four visible margins (*Margo interosseus*, *Margo cranialis*, *Margo caudalis*, *Margo lateralis*).

The distal end of the fibula forms the lateral malleolus (*Malleolus lateralis*) (Figure 8). Its medial surface bears a bony spicule for the fibular notch of the tibia and an articular surface (*Facies articulares malleoli*) for articulation with the calcaneus (*Calcaneus*).

The radiographic investigation of the crural region allowed the identification of the articular elements of the proximal and distal ends of the tibia and fibula and facilitated the identification of the tibiofibular interosseous space (Figure 6 and Figure 11), but it was difficult to identify small structures such as the malleolar groove.

The pes skeleton (*Skeleton pedis*) (Figure 9, Figure 10 and Figure 11) consists of the tarsal bones (*Ossa tarsi*), metatarsal bones (*Ossa metatarsalia*), phalanges of the pelvic limb (*Ossa digitorum pedis*), and the sesamoid bones (*Ossa sesamoidea*).

The tarsal area (*Regio tarsi*) showed us seven short bones, which form three rows. The proximal row has only two bones, the calcaneus (*Calcaneus*) laterally and the talus (*Talus*) medially. The middle row contains the central tarsal bone (*Os tarsi centrale* or *Os naviculare*) medially and the proximal half of the fourth tarsal bone (*Os tarsale IV* [*Os cuboideum*]) laterally. In a medio-lateral direction, the first tarsal bone (*Os tarsale I* [*Os cuneiforme mediale*]), second tarsal bone (*Os tarsale II* [*Os cuneiforme intermedium*]), third tarsal bone (*Os tarsale III* [*Os cuneiforme laterale*]), and the distal half of the fourth tarsal bone (*Os tarsale IV* [*Os cuboideum*]), constitute the distal tarsal row.

For tarsals, our investigation focuses only on the large first row elements.

The calcaneus (Figure 9, Figure 10 and Figure 11) is a long laterally located tarsal bone, which on its dorso-medial distal half has three articular surfaces: one in medial position, on the inner surface of the *sustentaculum tali*—the medial talar articular surface, the second one in a dorsal-lateral position—the proximal talar articular surface, and one close to the distal end—the distal talar articular surface for articulation with the talus (*Facies articulares talares*) (Figure 10). The calcaneal sulcus (*Sulcus calcanei*) (Figure 10) was discreet and located between the middle talar articular surface and the distal talar articular surface. The distal end presents a circular articular surface for the proximal extremity of the fourth tarsal bone (*Facies articularis cuboidea*) (Figure 10). The proximal half is well-developed and shows a rectangular aspect. The proximal end of it has a tuberous aspect and forms the calcaneal tuberosity (*Tuber calcanei*). Both the medial process (*Processus medialis*) and the lateral process (*Processus lateralis*) of the calcaneal tuberosity could be observed (Figure 10).

Medially, the calcaneal bone sends a bony projection that articulates with the plantar surface of the talus—this is the *sustentaculum tali*, and on the plantar surface of it, a tendinous groove (*Sulcus tendinis m. flexoris digit. lateralis*) was present. In an opposite position, on the lateral surface of the calcaneal bone, the peroneal tubercle is present (Figure 10).

The talus (Figure 9, Figure 10 and Figure 11) is shorter than the calcaneus (Table 8) and in a proximo-distal direction can be divided into three distinct segments: the body of the talus (*Corpus tali*), the neck of the talus (*Collum tali*), and the head of the talus *(Caput tali*) (Figure 10). The body represents approximately 50% out by the total length of the talus (Table 8) and is well-developed. Its proximal extremity bears the proximal trochlea (*Trochlea tali*) (Figure 10), which is part of the talocrural joint.

The medial surface of the body showed an articular surface (*Facies articularis malleolaris medialis*) (Figure 10) for articulation with the medial malleolus of the tibia. In a proximal position, a delicate medial tubercle (*Tuberculum medial*) (Figure 10) can be noticed. The proximal–lateral surface of the talus permitted the identification of a conical lateral tubercle (*Tuberculum lateralis*) (Figure 10). The proximal calcaneal articular surface (*Facies articulares calcaneae proximalis*) (Figure 10) and the middle calcaneal articular surface (*Facies articulares calcaneae media*) (Figure 10) covers the plantar aspect of the proximal end of the talus and articulate with the proximal and distal articular surfaces of the calcaneus. A deep and narrow *Sulcus tali* (Figure 10) passes between these two articular surfaces on the plantar aspect of the talus. In all specimens, the proximal surface was bigger and concave and had a proximo-lateral location compared with the middle one which was flat and had a proximal–medial position.

As only some of the specimens were available, a reduced set of metrical data were retrievable (Table 8). Data recorded (with a very limited set of available measurements due to a lack of pieces or the destruction of the specimens) show no significant variation among sexes.

The neck of the talus is a short and narrow segment, which continues distally in the body. The head has a tuberous aspect, and its distal end is covered by an articular surface for the central tarsal bone (*Facies articularis navicularis*) (Figure 10). The lateral surface of this segment carries a small distal articular calcaneal surface (*Facies articulares calcaneae distalis*) (Figure 10) for the distal articular talar surface of the calcaneal bone.

The metatarsal area has five metatarsal bones (*Ossa metatarsalia I–V*) (Figure 9 and Figure 11). Four of them have a similar anatomy, and the first one is shorter. Each metatarsal bone has a proximal base (*Basis*) for articulation with the distal row of the tarsal bones, a cylindrical shaft (*Corpus*) and a distal ball-shaped head (*Caput*) for the metatarsophalangeal joint (*Articulationes metatarsophalangeae*) (Figure 9).

Five digits are present (Figure 9 and Figure 11). The second, third, fourth, and fifth digits have a comparable size. The proximal phalanx (*Phalanx proximalis* [*Os compedale*]), middle phalanx (*Phalanx media* [*Os coronale*]), and the distal phalanx (*Phalanx distalis* [*Os unguiculare*, *Os ungulare*]) were well-developed in the last four digits.

For the pes region, the radiographic investigations (Figure 11) allowed the identification of all structures of the tarsal, metatarsal, and digital areas. The most notable features confirmed by the radiological investigation of these areas were related to the anatomical disposition of the tarsal bones at the level of the three tarsal rows. Also, the presence of the proximal sesamoid bones on the plantar side of all metatarsophalangeal joints of all digits (I–V) and the placement of the distal sesamoid bone on the plantar aspect of the distal interphalangeal joint of the first digit (I) could be noticed.

## 4. Discussion

The skeleton of the pelvic limb in African green monkeys revealed similar regions to those reported in domestic mammals by the classical anatomy books [43,44,45,46], and in Old World monkeys, New World monkeys, and humans [1,13,47,48,49]. As in other domestic mammalian species and other species of primates, in African green monkeys, the coxae bones have three distinguished segments: the ilium, ischium and pubis.

The anatomo-topographic position of this three-dimensional bony complex impacts the locomotor behavior, posture, and the parturition act [3,13,50,51]. Like other non-human primate species and humans, in African green monkeys or vervet monkeys the junction of the medial surfaces of the ilium with the articular surfaces of the sacrum forms the pelvic cingulum [1]. The length of the pubic symphysis and the anatomical aspects of the wing of the ilium can be used to differentiate between the monkeys and the apes. A longer pubic symphysis has been reported for monkeys compared with the apes, and the iliac wing was wider in apes than in monkeys [1,52,53] with a large variability in different orders of monkeys. The morphological features of the pubic symphysis in the studied specimens confirm the above statements. Similar aspects have been reported in *Sapajus libidinosus*, and a shorter pubic symphysis has been reported in sanguis and *Tarsius* [1,54].

The tridimensional shape of the coxae bone has a direct correlation with body size and locomotor behavior. According to recent studies, in strepsirrhines and haplorrhine primates, the width of the iliac wing increases with body weight [13,52]. This feature is key in visceral support during erect or semi-erect locomotion [1,55].

In *C. sabaeus*, the birth canal has a horizontal and cranio-caudal direction, with parallel planes between the pelvic inlet and pelvic outlet, and is typical for quadrupedal locomotion. In humans, with a bipedal type of pelvis, the birth canal is twisted in the middle segment inducing fetal rotation during parturition [55,56].

The acetabular cavity in African green monkeys was large and well-developed. Similar features of the acetabulum, doubled by a short pubic symphysis have been reported for the vertical clingers and leapers. Opposite, in arboreal quadrupedal *strepsirrhines*, a reduced acetabulum and a long pubic symphysis have been reported [13,57].

In *C. sabaeus*, the femur bone was the longest bone of the pelvic limb, similar to the reported data for *S. libidinosus* and humans [1]. Even if in the common marmoset [58] and the Goeldi’s marmoset [59] the femur was well developed, this bone was not the longest bone of the pelvic limb. The anatomical segments of the femur described by us for the studied specimens were similar to the data reported in Mangabeys and Guenons [60], *Papio hamadryas* [58], *S. libidinosus* and humans [1,61].

The femoral head was spherical for the AGM (*C.sabaeus*) specimens, similar to the cebid monkeys. A hemispherical femoral head has been reported for the *Papio* genus monkeys and the African mangabey monkeys [60]. Similar to the black-crested Sumatran langur monkeys and most cercopithecines in *C. sabaeus*, the *fovea capitis* has an inferior location reported to the central area of the femoral head [57]. According to [62], the anatomical position of the fovea capitis has a direct relation with the anatomical position of the femur during postural and locomotor activity, and the depth of it can be related to the size of the ligament of the femoral head.

A well-developed greater tubercle, which extends proximally to the femoral head, was observed in *C. sabaeus*. Similar aspects were reported for the Hamadryas Baboon (*Papio hamadryas*) [63], *Macaca mulatta* [48], *Lemur catta* [54], *Cercocebus torquatus*, *Cercocebus galeritus*, *Cercopithecus mitis*, *Cercopithecus mona*, *Cercopithecus aethiops*, *Papio hamadryas*, *Colobus guereza*, *Presbytis melalophos*, *Nasalis Larvatus*, and *Pan troglodytes* [60]. In *S. libidinosus* [1], *Alouatta seniculus*, *Cebus capucinus*, *Ateles paniscus* [60], and humans [37,38] the apex of the greater trochanter ends almost at the same level as the femoral head. This bony prominence provides insertion points for the *gluteus maximus*, *gluteus medius*, *gluteus profundus*, *piriformis*, internal obturator, and the *gemelli* muscles [48,63]. A tall greater trochanter helps the muscles to develop a higher force and limits the abduction movements of the hip joint, while a shorter trochanter increases the coxofemoral joint mobility. According to [60], in *cercopithecines* and *colobine* monkeys, due to the anatomic aspect of the greater trochanter, which is higher than the femoral head, the mobility of the coxofemoral joint has some limitations. Similar to other primates [37,48,63], in *C. sabaeus* the presence of the lesser trochanter could be confirmed for all the specimens. Refs. [54,64] reported the absence of the intertrochanteric crest in *Lemur catta*, between the greater trochanter and the lesser trochanter—a well-developed structure in our specimens.

The absence of the third trochanter that we reported in the African green monkeys is similar to the reported data for baboons, rhesus monkeys [48,63], *S. libidinosus*, and *A. seniculus* [1]. In healthy ring-tailed lemurs (*Lemur catta*), most *strepsirrhines* and *callitrichids* [54], the same structure is well developed and has a distal position reported to the lesser trochanter. The reported data show a huge variability in this structure in humans [61,65,66]. The results pointing to the presence of the intertrochanteric line in African green monkeys were similar to the data reported in the bearded capuchin monkeys [1] and different from the reported data in the common marmoset [47], Venezuelan red howler monkey [67], and the Goeldi’s marmoset [68]. As in humans [38,65] and other primates [60,67], the gluteal tuberosity of the *C. sabaeus* was represented by a small bony prominence. In diademed sifaka monkeys and Senegal bushbaby primates, the same structure was robust and well-developed [1,4,69].

The anatomical structures, which are part of the distal epiphysis of the femur in our specimens showed similar anatomic aspects with reported data in baboons [1], *S. libidinosus* [1], and other mangabeys, and guenon monkeys [60]. In the studied specimens, similar to the *Papio hamadryas* and the *S. libidinosus* [1,63], the medial condyle of the femur was better developed than the lateral one. According to [70], this aspect shows that the medial condyle bears and transmits more body weight compared to the lateral condyle. Also, it can suggest that during the locomotion, the stifle joint realizes abduction movements. The same authors suggest that in the mangabey and guenon species, the medial condyle of the femur is 12% to 22% bigger than the lateral condyle [60].

In the jumping species of monkey, the femoral condyles have very similar dimensions, while in the arboreal quadrupedal monkeys, the medial condyle is wider compared with the lateral condyle [57].

The presence of the extensor fossa at the level of the distal epiphysis, confirms the origin of the long digital extensor muscle at this level [71]. In *Lemur catta*, the cranial surface of the head of the fibula, the lateral condyle of the tibia, and the interosseous membrane, represent the origin points for this muscle, and this is the reason why the femur does not have an extensor fossa [54,72].

In *C. sabaeus*, the patella was the largest sesamoid bone and the anatomical aspects of it are very similar to the data reported for humans [61,73] and other primates [1,47,58,68]. The gastrocnemius sesamoid bones (*Ossa sesamoidea m. gastrocnemii)* were easily identifiable on the radiograph performed as double small bony structures located behind the lateral and medial femoral condyles [35,43,44,45]. There was no indication of the existence of popliteal sesamoid bones (*Os sesamoideum m. popliteii)* in the investigated specimen. This is mainly because of the limited choice of studied bones from the osteological collection and the limitation given by only one radiologically investigated individual and does not exclude the existence of this particular bony element.

The tibia, in African green monkeys, was shorter compared with the femur; the same features were reported for some monkey species [1,46,61] and in the classic anatomy books for non-primates mammal species [43,46,74]. A longer tibial bone has been reported for the common marmoset and the Goeldi’s marmoset [47,68].

Similar to the reported data in *Macaca mulatta*, *S. libidinosus*, and *Papio hamadryas* [1,63,75] in our specimens, the proximal end of the tibia had a triangular aspect and was covered by a medial and a lateral articular surface, with the medial one larger than the lateral one. According to [76], this feature is characteristic of anthropoid primates.

This area is very important for the stifle joint biomechanics, being the place where osteoarthritic lesions occur [77]. The concave features of the articular surfaces and the cranio-caudal direction of their longitudinal axis can be correlated with the flexion and extension movements of the stifle joint [76].

The presence of the medial and lateral intercondylar eminences, reported by us in *C. sabaeus*, was also reported previously in *Papio hamadryas* and *S. libidinosus* [1,63]. A simple, intercondylar eminence has been reported in *Lemur catta* [54] and *Tarsius* [76], and the absence of it in *Macaca mulatta* [54,75,76].

The presence of a plane articular surface on the caudal–ventral surface of the lateral tibial condyle, for the fibular head, reported by us in African green monkeys, was also reported in rhesus monkeys [75] and other primates [48,63], but a convex aspect of the same structure has been reported in Lemur catta monkeys [1,54]. In humans, the proximal tibio-fibilar joint has a lateral disposition at the level of the lateral condyle of the tibia [78], and the head of the fibula bears a styloid process [79], which has not been seen in *Chlorocebus sabaeus*.

The fibula was longer than the tibia (Table 7), and the interosseous space was visible between these two bones. This feature has been reported also in humans [37,73] and other primates [9,48,68,75]. According to [1,9,76] in *Tarsius*, *Afrotarsius*, and *Necrolemur*, the fibula fuses with the tibia up to the middle third of it, with the interosseous space being shorter for these species. Individualized and unfused tibia and fibula have been reported for *Parapithecids* and other anthropoids [57,80]. The macroscopic features of the tibio–fibular complex provide valuable information related to the locomotor behavior of distinct species. In *Tarsiers*, and other jumping mammals, a partial fusion at the level of the distal end of the bones has been reported. Proximal and distal tibio–fibular fusions have been reported for burrowing mammals like mole rats and armadillos. Special features were reported in running species like ungulates where the fibula is fused with tibia and only the distal end of it forms a detachable bone, the malleolar bone. In all the above situations, fixed fibulas have been described. The same authors reported the presence of the mobile type of fibula in carnivores and primates [9,10,12].

Even if in humans, during locomotion, the fibula is considered to have a rudimentary role, there are a lot of studies that confirm that this bone can bear 6.4–19% of the total body weight during plantigrade locomotion [9,81,82,83]. This action impacts the robusticity of the fibula. Based on some reported data, Ref. [9] concluded that the cercopithecoid primates have a less robust fibula than the hominoids.

The skeletal elements of the tarsal area in *C. sabaeus*, are formed by seven bones disposed in three distinctive rows. The proximal row includes the talus and calcaneus, the middle row the navicular bone, the proximal half of the fourth carpal bone and the distal row of tarsal bones, including the first, second, third, and fourth tarsal bones. A similar number of tarsal bones and a similar distribution of them has been reported in *S. libidinosus* [1], *Papio hamadryas* [63], *Lemur catta* [54,64], common marmoset [47], *C. goeldii* [68], and humans [73].

The bones of the first tarsal bones are relevant for different research areas like primatology, anthropology, archaeozoology, vertebrate paleontology, phylogeny, and taxonomy [84,85,86,87,88,89,90]. These two bones were used in morphometric studies, which correlated their size and body mass. Ref. [91] in his study showed that for land mammals the tibial tarsal bone (talus, astragalus) provides the most accurate information related to the body mass. In primates, both bones, the tibial tarsal bone and the fibular tarsal bone, can be used for body mass estimation.

Distally from the tarsal area, the anatomical features of the skeletal structures located in the metatarsal area and the digit area were similar to data reported for other quadrupedal primates [47,58]. The presence of the proximal sesamoid bones is noted for all digits, and the existence of the distal sesamoids was highlighted in the case of digit I structures, similar to data published in the case of lemurs and white-footed tamarins [54,92,93].

## 5. Conclusions

Providing a precise and accurate description of the pelvic limb skeleton in *Chlorocebus sabaeus*, this study adds additional information regarding the bone anatomy of these monkeys. With so many details, the present study can be used by medical researchers as a starting point in areas like skeletal malformations, bone pathologies, orthopedics, and traumatology, surgery, skeletal morphophysiology, primatology, and paleontology. Because vervet monkeys (*C*. *sabaeus*) are widespread in many zoos, this study can be helpful for practitioners who deal with exotic animals.

## Figures and Tables

**Figure 1 animals-15-00209-f001:**
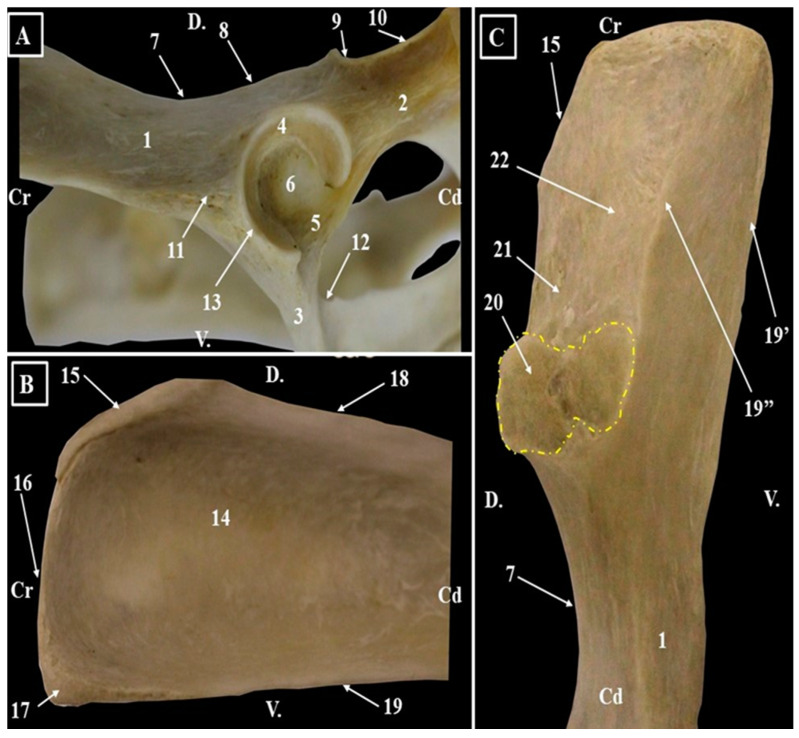
Anatomical features of the ilium and acetabulum (**A**). Acetabulum; (**B**). Iliac wing—lateral surface; (**C**). Iliac wing-medial surface. 1. Ilium; 2. Ischium; 3. Pubis; 4. Lunate surface; 5. Acetabular notch; 6. Acetabularossa; 7. Greater ischiatic notch; 8. Ischiatic spine; 9. Tendinous groove; 10. Lesser ischiatic notch; 11. Lateral area of the rectus femoris muscle; 12. Obturator groove; 13. Acetabular margin; 14. Gluteal fossa; 15. Sacral tuberosity of ilium; 16. Crest of ilium; 17. Coxal tuberosity; 18. Dorsal iliac spine; 19. Ventral iliac spine; 19′. Outer lip of the ventral iliac spine; 19″. Inner lip of the ventral iliac spine; 20. Auricular surface; 21. Iliac tuberosity; 22. Iliac fossa.

**Figure 2 animals-15-00209-f002:**
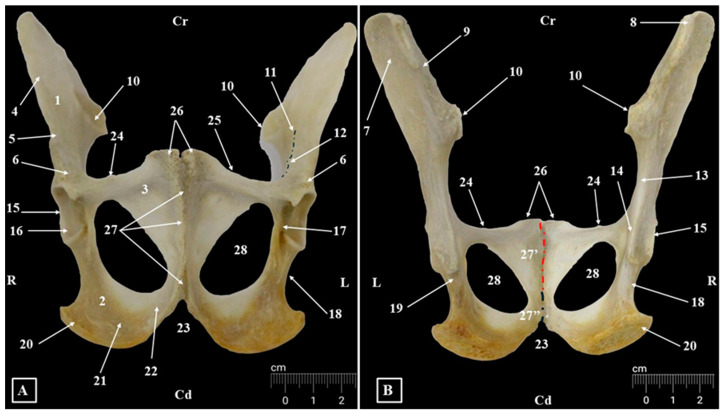
Anatomical features of the coxae bones (**A**). Ventral view; (**B**). Dorsal view. 1. Ilium; 2. Ischium; 3. Pubis; 4. Wing of ilium; 5. Body of ilium; 6. Lateral area of the rectus femoris muscle; 7. Gluteal fossa; 8. Sacraltuberosity; 9. Dorsal iliac spine; 10. Auricular surfaces; 11. Arcuate line; 12. Psoas minor tuberosity; 13. Greater ischiatic notch; 14. Ischiatic spine; 15. Acetabular margin; 16. Facies lunata; 17. Acetabular notch; 18. Lesser ischiatic notch; 19. Tendinous groove; 20. Ischial tuberosity; 21. Ischial body; 22. Ramus of ischium; 23. Ischial arch; 24. Iliopubic eminence; 25. Pecten pubis; 26. Pubic tubercle; 27. Pelvic symphyses; 27′. Pubic symphysis; 27″. Ischial symphysis; 28. Obturator foramen.

**Figure 3 animals-15-00209-f003:**
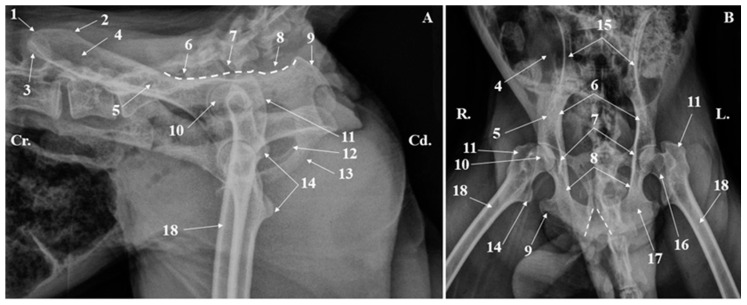
Coxae bones radiograph (**A**). Lateral–lateral projection; (**B**). Ventrodorsal projection. 1. Iliac crest; 2. Sacraltuberosity; 3. Coxal tuberosity; 4. Wing of the ilium; 5. Body of the ilium; 6. Greater ischiatic notch; 7. Ischiatic spine; 8. Lesser ischiatic notch; 9. Ischiatic tuberosity; 10. Femoral head; 11. Greater trochanter; 12. Obturator foramen; 13. Pelvic symphysis; 14. Lesser trochanter; 15. Ilium; 16. Femoral neck; 17. Ischium; 18. Femoral shaft. V-shape dotted lines-the ischiatic arch.

**Figure 4 animals-15-00209-f004:**
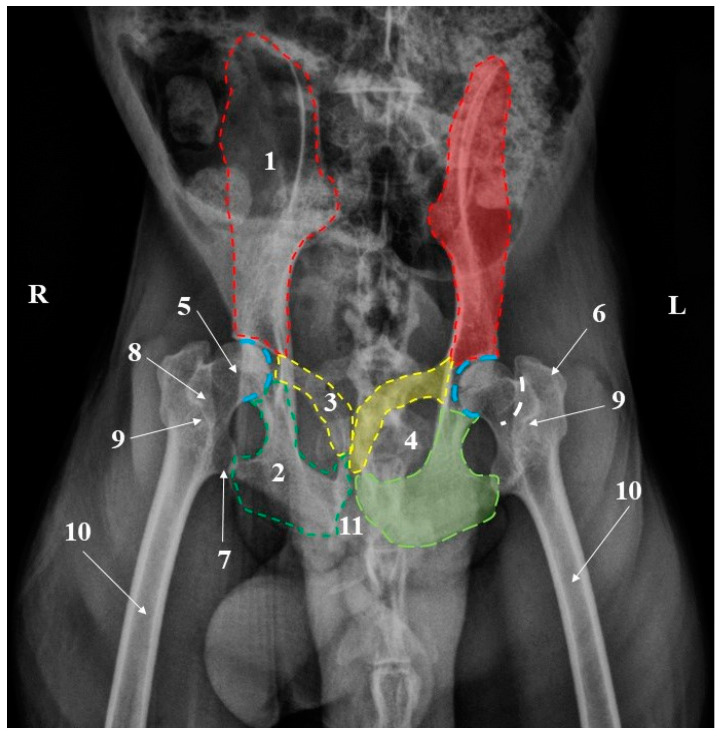
Pelvic girdle radiograph-ventrodorsal projection. 1. Ilium; 2. Ischium; 3. Pubis; 4. Foramen ovale; 5. Femoral head; 6. Greater trochanter; 7. Lesser trochanter; 8. Trochanteric fossa; 9. Intertrochanteric crest; 10. Femoral shaft; 11. Ischiatic arch. Blue dotted line-the acetabulum; White dotted line-the femoral neck.

**Figure 5 animals-15-00209-f005:**
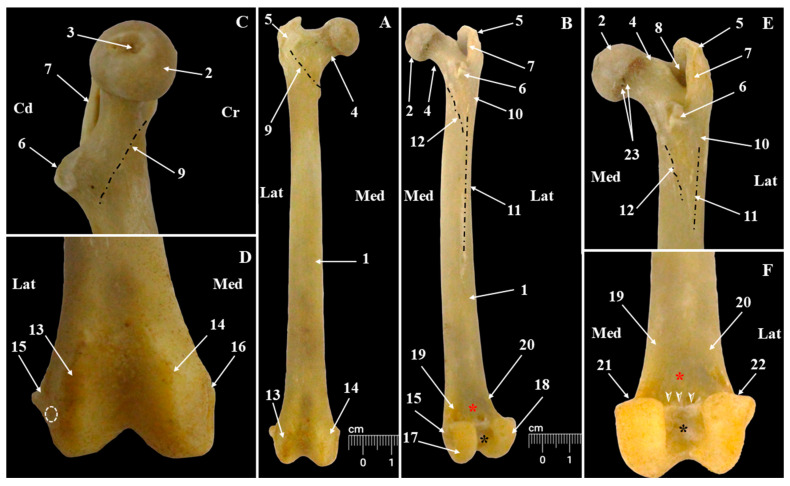
Anatomical features of the femur. Cranial aspect (**A**),Caudal aspect (**B**), Details of the medial part of proximal extremity (**C**), Details of the cranial part of distal extremity (**D**), Details of the caudal part of proximal extremity(**E**), Femur- details of the caudal part of distal extremity (**F**) 1. Femoral shaft; 2. Femoral head; 3. Fovea for ligament of head of femur; 4. Neck of femur; 5. Greater trochanter; 6. Lesser trochanter; 7. Intertrochanteric crest; 8. Trochanteric fossa; 9. Intertrochanteric line; 10. Gluteal tuberosity; 11. Linea aspera; 12. Pectineal line of femur; 13. Lateral trochlear lip; 14. Medial trochlear lip; 15. Lateral epicondyle; 16. Medial epicondyle; 17. Medial femoral condyle; 18. Lateral femoral condyle; 19. Medial supracondylar line; 20. Lateral supracondylar line; 21. Articular surface for the medial gastrocnemius sesamoid bone; 22. Articular surface for the lateralsesamoid bone of gastrocnemius; 23. Vascular foramina; Black asterix—Intercondylar fossa; White arrowheads—intercondylar line; Red asterix—popliteal fossa. Dotted circle (**D**)—ligamentary fossa.

**Figure 6 animals-15-00209-f006:**
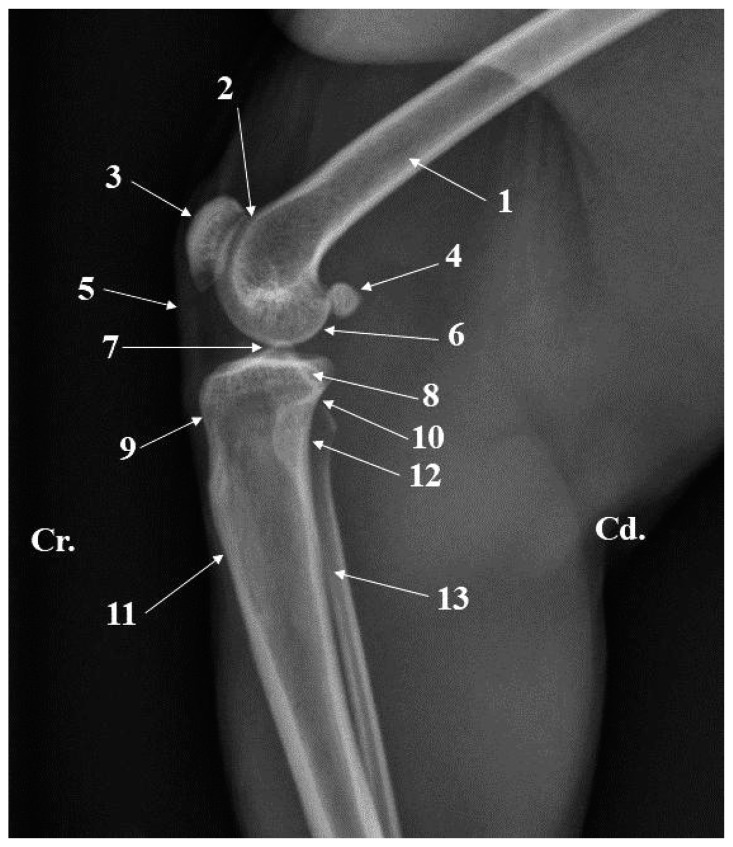
Stifle joint radiograph—medio-lateral projection. 1. Femur; 2. Femoral trochlea; 3. Patella; 4. Sesamoid bone of gastrocnemius; 5. Patellar ligament; 6. Femoral condyle; 7. Intercondylar tuberosity; 8. Medial condyle of the tibia; 9. Tibial tuberosity; 10. Proximal tibio-fibilar joint; 11. Tibial tuberosity; 12. Head of fibula; 13. Body of fibula.

**Figure 7 animals-15-00209-f007:**
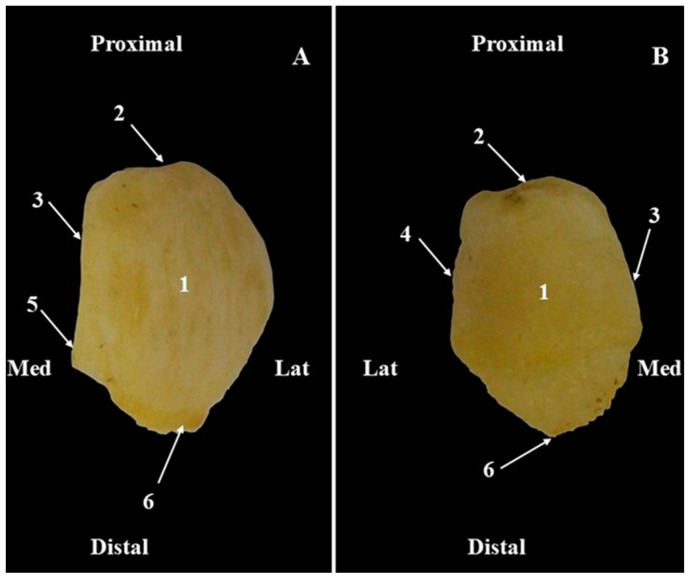
Anatomical features of the patella (**A**). Cranial surface; (**B**). Articular surface. 1. Patella; 2. Base of patella; 3. Medial margin of the patella; 4. Lateral margin of the patella; 5. Medial patellar tuberosity; 6. Apex of the patella.

**Figure 8 animals-15-00209-f008:**
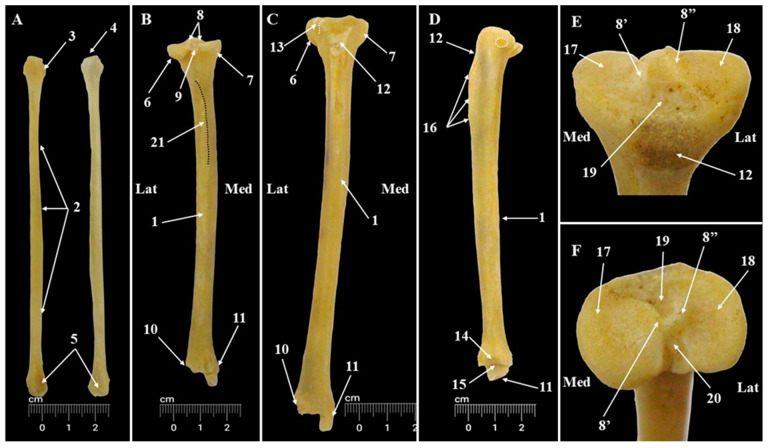
Anatomical features of the tibia and fibula bones (**A**). Fibula (lateral-left and medial-right); (**B**). Tibia caudal view; (**C**). Tibia cranial view; (**D**). Tibia lateral view; (**E**,**F**). Proximal end of tibia. 1. Body of tibia; 2. Body of fibula; 3. Head of fibula; 4. Articular surface of head of fibula; 5. Lateral malleolus; 6. Lateral tibial condyle; 7. Medial tibial condyle; 8. Intercondylar eminence; 8′. Medial intercondylar tubercle; 8″. Lateral intercondylar tubercle; 9. Popliteal notch; 10. Fibular notch; 11. Medial malleolus; 12. Tibial tuberosity; 13. Extensor’s groove; 14. Distal fibular articular surface; 15. Tibial cochlea; 16. Tibial tuberosity; 17. Medial condyle; 18. Lateral condyle; 19. Cranial intercondylar area; 20. Caudal intercondylar area.

**Figure 9 animals-15-00209-f009:**
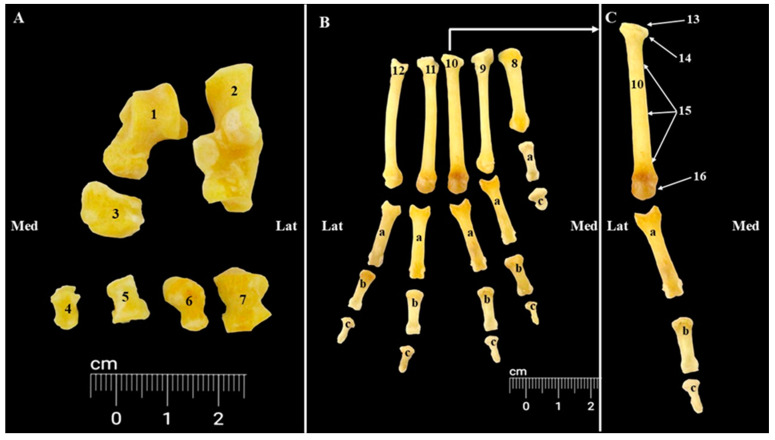
Anatomical features of the pes region (**A**). Tarsal bones—dorsal view; (**B**). Digital area—dorsal view; (**C**). Third digit skeleton—dorsal view. 1. Talus; 2. Calcaneus; 3. Central tarsal bone; 4. First tarsal bone; 5. Second tarsal bone; 6. Third tarsal bone; 7. Fourth tarsal bone; 8. First metatarsal bone; 9. Second metatarsal bone; 10. Third metatarsal bone; 11. Fourth metatarsal bone; 12. Fifth metatarsal bone; 13. Articular surface for distal tarsal bone; 14. Base of metatarsal bone; 15. Body of metatarsal bone; 16. Head of metatarsal bone; a. First phalanges of digits I–V; b. Second phalanges of digits II–V; c. Third phalanges of digits I–V.

**Figure 10 animals-15-00209-f010:**
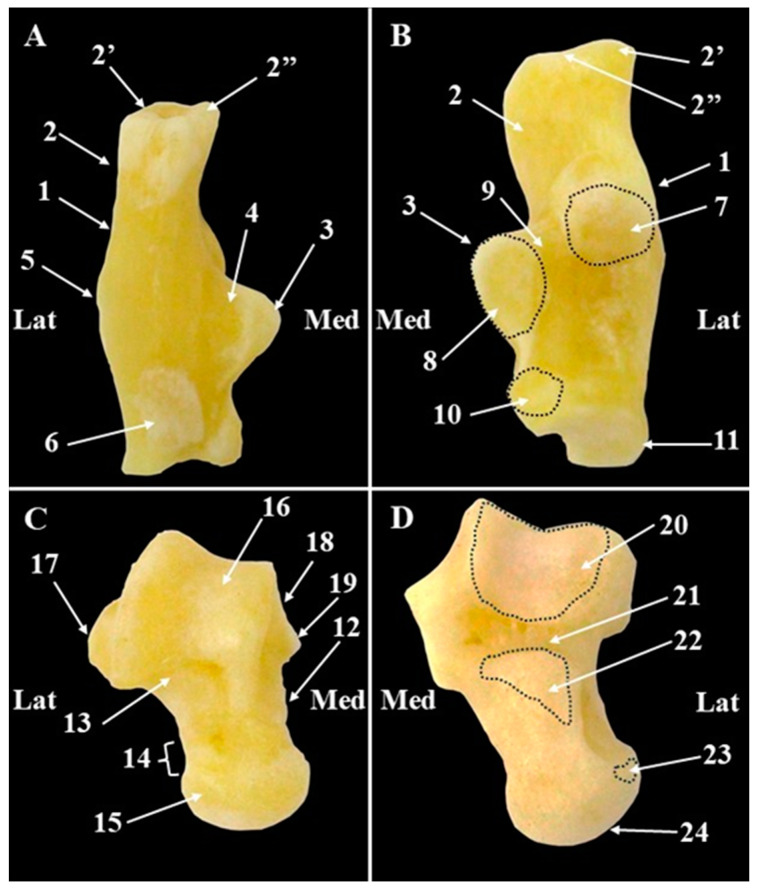
Anatomical features of the calcaneus and talus bones (**A**). Calcaneus-plantar view; (**B**). Calcaneus—dorsal view; (**C**). Talus—dorsal view; (**D**). Talus—plantar view. 1. Calcaneus; 2. Calcaneal tuberosity; 2′. Lateral calcaneal process; 2″. Medial calcaneal process; 3. Sustentaculum tali; 4. Tendinous groove; 5. Peroneal tubercle; 6. Calcaneal tubercle; 7. Proximal talar articular surface; 8. Middle talar articular surface; 9. Calcaneal sulcus; 10. Distal talar articular surface; 11. Articular surface for the fourth tarsal bone; 12. Talus; 13. Body of the talus; 14. Neck of the talus; 15. Head of the talus; 16. Trochlea of the talus; 17. Lateral tubercle; 18. Medial talar articular surface; 19. Medial tubercle; 20. Proximal talar articular surface; 21. Talar sulcus; 22. Lateral talar articular surface; 23. Distal talar articular surface; 24. Articular surface for the central tarsal bone.

**Figure 11 animals-15-00209-f011:**
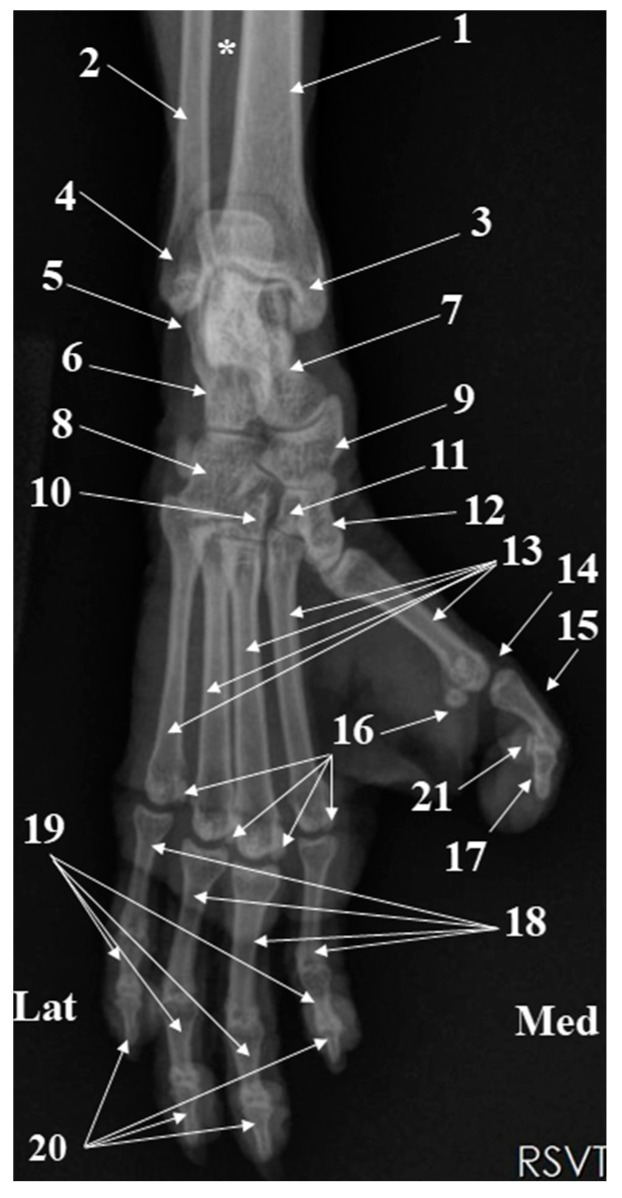
Pes area-radiograph dorso-plantar projection. 1. Tibia; 2. Fibula; 3. Medial malleolus; 4. Lateral malleolus; 5. Peroneal tubercle; 6. Calcaneus; 7. Talus; 8. Fourth tarsal bone; 9. Central tarsal bone; 10. Third tarsal bone; 11. Second tarsal bone; 12. First tarsal bone; 13. Metatarsal bone I–V; 14. Metacarpophalangeal joint of digit I; Proximal phalanx of digit I; 16. Proximal sesamoids; 17. Distal phalanx of digit I; 18. Proximal phalanges of digits II–V; 19. Middle phalanges of digits II–V; 20. Distal phalanges of digits II–V; 21. Distal sesamoid of digit I, Asterix—tibiofibular interosseous space.

**Table 1 animals-15-00209-t001:** Measurements for the pelvic bone. Codings and explanations on different measurements.

Measurement Code		Name of the Measurements
P1	GL	maximum length of the pelvis
P2/3	LA/LAP	length of the acetabulum
P4	LS	length of the symphysis
P5	SH	height of the iliac body
P6	SB	minimum diameter of the iliac body
P8	Lfc	internal length of the obturator foramen
P8A	smOF	smallest diameter of the obturator foramen
P9	GBT/ASIS width (as in)	greatest breadth between the iliac spines/pelvic width is measured between both anterior superior iliac spine (ASIS) apexes
P10	GBA	greatest breadth between the highest points of the acetabular cavities
P11	GBt	greatest breadth between the ischiatic tuberosities
P12	SBD	greatest breadth between the lesser sciatic notches from dorsal
P20		pelvis depth—the highest dorsoventral diameter of the iliac blades
P21	PSIS width	distance between the internal iliac angles/distances between the posterior superior iliac spines
P22	auricular area surface	measured with the free tool ImageJ (1.54k 15 September 2024)
P23	area of the obturator foramen	measured with the elliptic tool ImageJ (1.54k 15 September 2024)

**Table 2 animals-15-00209-t002:** Measurements for the femur. Codings and explanations on different measurements.

Measurement Code		Name of the Measurements
F1	GL/ML	maximum length head-medial condyle/natural femoral length (measured from the summit of the femoral head to the longest point of the medial condyle)
F2	GLC	greatest length at articular head/femoral shaft length (measured along the main axis of the shaft from the femoral head to the transversal line that joins the distal bicondylar area)
F3	Bp	proximal width
F5	DC/THD	cranio-caudal diameter of the articular head
F6	SD/T-MSD/Pi1	minimum transversal diameter of the diaphysis
F8	Bd	transversal diameter of the distal epiphysis
F10	AP-MSD	cranio-caudal diameter of the diaphysis
F11	TL	trochantero-condylar length (from the summit of the first trochanter to the transversal line that joins the distal bicondylar area)
F12	BCL	bicondylar length (maximal length head-bicondylar line)
F13A	VHD	vertical head diameter
F13B	HHd	horizontal head diameter
F14	BTL	bi-trochanteric distance
F15	ML-MC	maximum cranio-caudal length of the medial condyle
F16	ML-LC	maximum cranio-caudal length of the lateral condyle
F17	HAG/FA1	head angulation
F19	WMC	width of the medial condyle
F20	WLC	width of the lateral condyle
F21	APDistMed	cranio-caudal diameter of the medial part of the distal end (caudal condyle-cranial margin of the corresponding trochlear rim)
F22	APDistLat	antero-posterior diameter of the lateral part of the distal end (posterior condyle-anterior margin of the corresponding trochlear rim
F23	ST1/Pl1	cranio-caudal subtrochanteric width
F24	ST2/Pl2	medio-lateral subtrochanteric width
F25	Pi2	cranio-caudal diameter of the diaphysis (*at linea aspera*)
F26	Po1	cranio-caudal width above popliteal region
F27	Po2	latero-medial width above popliteal region
F28/F18	FNHL	femoral head and neck length (measured along the angulation line of the head)
F29	FNd	femoral neck diameter (smallest diameter)

**Table 3 animals-15-00209-t003:** Measurements for the tibia and fibula. Codings and explanations on different measurements.

Measurement Code		Name of the Measurements
T1	GL	maximum length (measured from tibial intercondylar eminence to the most distal point-medial malleolus)
T2	MLT	lateral length of the bone (measured from the lateral condyle to the lateral malleolus)
T3	OLT	oblique length of the tibia (measured from the lateral condyle to the medial malleolus)
T4	TL	medial length of the bone (measured from the medial condyle to the fibular notch)
T5	APWTP	cranio-caudal width of the tibial plateau
T6	MLWPT	maximum latero-medial diameter of the proximal epiphysis (tibial plateau)
T7	CTP	circumference of the area of the tibial plateau
T8	ASAS	area of the tibial plateau
T9	MLMC	maximal length of the medial condyle
T10	MLLC	maximal length of the lateral condyle
T11	APUTh	cranio-caudal diameter of the upper third of the shaft
T12	LMUth	latero-medial diameter of the upper third of the shaft
T13	APD	antero-posterior diameter of the diaphysis
T14	LMD	latero-medial diameter of the diaphysis
T15	MWMC	maximal width of the medial condyle
T16	MWLC	maximal width of the lateral condyle
T17	TWDT	transversal width of the distal epiphysis
T18	APDT	cranio-caudal length of the distal epiphysis
Fi1	GL	maximum length

**Table 4 animals-15-00209-t004:** Measurements talus and calcaneus. Codings and explanations on different measurements.

Measurement Code		Name of the Measurements
Ta1	TaL	talar length
Ta2	TaW	talar width
Ca1	MaxL	maximum length calcaneus -the linear distance between the most cranial point of the calcaneus and the most caudal point on the calcaneal tuberosity
Ca2	MaxH	maximal height of sustentaculum—the distance between the most proximal and the most distal points on the calcaneal tuberosity

**Table 5 animals-15-00209-t005:** Available metrical data for the pelvis (mm).

Measurement Code	MM	A438	K920	K945	K930	V585	Mean Values for Males	Mean Values for Females
P1	GL	12.72	12.08	12.27	13.03	11.69	12.460	12.205
P2/3	LA/LAP	1.51	1.7	2.01	1.81	1.62	1.840	1.565
P4	LS	4.55	3.7	3.1	2.28	3.57	3.027	4.060
P5	SH	1.47	1.37	1.26	1.01	1.17	1.213	1.320
P6	SB	1.11						
P8	Lfc	3.2	2.66	2.85	2.55	2.48	2.687	2.840
P8a	sdOF	1.16	1.81	1.9	1.91	1.73	1.873	1.445
P9	GBT/ASIS width (as in)	11.6	11.1	-	-	-	11.100	11.600
P10	GBA	8.16	8.51	-	-	-	8.510	8.160
P11	GBt	7.41	8.04	-	-	-	8.040	7.410
P12	SBD	6.2	7	-	-	-	7.000	6.200
P20		2.88	2.67	3.08	3.07	2.57	2.940	2.725
P21	PSIS width	4.93		-				4.930
P22	auricular area surface	1.523	0.9134		1.281	1.357	1.097	1.440
P23	area of the obturatory foramen (mm^2^)	4.21	3.694	4.62	4.07	3.16	4.128	3.685

**Table 6 animals-15-00209-t006:** Available metrical data for the femur (mm).

		A438-R	A438-L	K920-R	K920-L	K930-R	K930-L	V585-R	V585-L	K945-R	K945-L	Mean Values for Males	Mean Values for Females
F1	GL/ML	14.14	14.58	12.14	13.38	16.74	16.6	13.85	13.5	16.19	16.5	15.377	13.675
F2	GLC	13.96	14.6	12.24	13.61	16.66	16.7	13.34	13.46	16	16.4	15.268	13.400
F3	Bp	2.57	2.21	2.69	2.63	3.31	3.4	2.51	3.05	3.05	2.93	3.002	2.780
F5 = F13B	DC/THD	1.26	1.28	1.34	1.35	1.43	1.46	1.29	1.27	1.37	1.36	1.385	1.280
F6	SD/T-MSD/Pi1	1.07	1.1	0.88	1	1.17	1.1	1	0.92	1.08	1.07	1.050	0.960
F8	Bd	2.36	2.38	1.92	2.31	2.71	2.74	2.47	2.28	2.6	2.66	2.490	2.375
F10	AP-MSD	0.96	0.94	0.9	0.89	1.02	1.02	0.83	0.84	0.98	0.92	0.955	0.835
F11	TL	13.72	14.76	12.4	13.51	16.86	17	13.58	13.69	16.4	16.6	15.462	13.635
F12	BCL	14.06	14.55	12.14	13.5	16.85	16.59	13.49	13.38	15.94	16.4	15.237	13.435
F13A	VHD	1.24	1.28	1.29	2.2	1.4	1.52	1.2	1.21	1.27	1.32	1.500	1.205
F13B	HHd	1.31	1.37	1.49	1.65	-	1.72	1.27	1.29	1.45	1.5	1.562	1.280
F14	BTL	2.86	2.39	2.02	2.2	3.16	2.68	2.35	2.41	2.85	2.93	2.640	2.380
F15	ML-MC	1.28	1.34	1.14	1.23	1.25	1.33	1.18	0.99	1.16	1.26	1.228	1.085
F16	ML-LC	1.27	1.39	1.04	0.94	1.11	1.19	1.06	1.01	1.09	1.05	1.070	1.035
F17	HAG/FA1	47.9	46.98	65	69	58.55	64.5	64.1	61.5	58	51.5	61.092	62.800
F19	WMC	0.69	0.81	0.64	0.84	1.1	1.03	1.14	0.85	0.9	1.03	0.923	0.995
F20	WLC	0.65	0.64	0.55	0.69	0.86	1.16	0.77	0.74	0.81	0.8	0.812	0.755
F21	APDistMed	1.93	2.01	2	1.95	2.05	2.21	1.94	1.85	1.77	1.99	1.995	1.895
F22	APDistLat	2.21	2.35	1.89	1.87	2.11	2.18	1.88	1.86	1.97	1.96	1.997	1.870
F23	ST1/Pl1	1.05	1.1	0.91	1.1	1	1.1	0.88	0.93	1.17	-	1.056	0.905
F24	ST2/Pl2	1.12	1.25	0.98	0.94	1.29	1.29	0.97	0.96	1.17	1.08	1.125	0.965
F25	Pi2	1.01	0.98	0.73	0.89	1.05	1.07	0.91	0.83	1.06	0.95	0.958	0.870
F26	Po1	1.12	1.02	0.94	0.92	1.39	1.5	0.99	0.98	1.01	1.08	1.140	0.985
F27	Po2	1.41	1.31	1.09	1.25	1.03	1.06	1.13	1.2	1.4	1.32	1.192	1.165
F28/F18	FNHL	2.9	3.02	2.74	2.6	3.42	3.42	2.81	2.73	2.92	2.75	2.975	2.770
F29	FNd	0.81	0.8	0.92	0.98	1.11	1.29	0.95	0.95	0.95	1.04	1.048	0.950
Diaphyseal Index (F1/F6 Ratio)		7.57	7.54	7.25	7.47	6.99	6.63	7.22	6.81	6.39	6.48	6.869	7.018
Proximal Epiphyseal Index (F1/F3 Ratio)		41.63	49.77	32.71	38.02	35.35	32.35	39.84	30.16	35.41	36.52	35.061	35.002
Distal Epiphyseal Index (F1/F8 Ratio)		45.34	46.22	45.83	43.29	43.17	40.15	40.49	40.35	41.54	40.23	42.368	40.418

**Table 7 animals-15-00209-t007:** Available metrical data for the tibia and fibula (mm).

	mm	A438 R	A438 L	K920 R	K920 L	K930 R	K930 L	K945 R	K 945 L	V585 L	Mean Values for Males	Mean Values for Females
T1	GL	13.8	13.42	15.57	14.91	16.56	16.6	14.74	15.73	13.1	15.69	13.44
T2	MLT	12.45	12.85	14.96	13.81	15.5	15.79	14.14	15.6	12.72	14.97	12.67
T3	OLT	12.82	13.12	15.67	14.56	16.3	16.47	14.67	15.62	13.21	15.55	13.05
T4	TL	12.42	13.59	15.53	14.59	15.62	15.58	14.14	15.14	12.99	15.10	13.00
T5	APWTP	1.7	1.8	1.85	-	2.18		1.9	2.03	1.83	1.99	1.78
T6	MLWPT	2.05	2.15	2.45	2.24	2.64	2.67	2.34	2.43	2.14	2.46	2.11
T7	CTP	6.349	6.318	6.847	-	8.54	-	7.92	6.787	7.156	7.52	6.61
T8	ASAS	1.82	1.924	2.078	-	3.4	-	2.582	2.857	1.961	2.73	1.90
T9	MLMC	1.18	1.07	1.06	-	1.51	-	1.28	1.26	1.13	1.28	1.13
T10	MLLC	1.09	1.03	0.99	-	1.30	-	1.34	1.32	0.99	1.24	1.04
T11	APUTh	-	1.25		1.26	1.42	1.4	1.15	1.2	1.13	1.29	1.19
T12	LMUth	0.93	0.89	0.91	0.84	0.92	0.97	0.87	0.99	0.85	0.92	0.89
T13	APD	0.94	0.97	0.96	1.11	0.98	1.07	0.86	0.94	0.72	0.99	0.88
T14	LMD	0.75	0.69	0.84	0.79	0.89	0.84	0.73	0.81	0.69	0.82	0.71
T15	MWMC	0.93	0.84	0.85	-	1.05	-	0.88	0.85	0.78	0.91	0.85
T16	MWLC	0.77	0.79	0.79	-	0.85	-	0.98	1.09	0.83	0.93	0.80
T17	TWDT	1.43	1.57	1.67	1.57	1.73	1.88	1.57	1.62	1.48	1.67	1.49
T18	APDT	-	1.38	-	1.69	2.05	1.67	1.45	1.45	1.39	1.66	1.39
Fi1	Gl	11.85	11.73	-	14.85	14.95	14.55	14.19	14.48	11.42	14.60	11.67

**Table 8 animals-15-00209-t008:** Available metrical data for the talus and calcaneus (mm).

	A438 R	A438 L	K920 R	K945 R	K 945 L	Mean Values for Males	Mean Values for Females
TAL	2.08	2.07	2	2.24	2.16	2.13	2.08
TaW	1.33	1.27	1.4	1.33	1.38	1.37	1.30
MAxL	2.9	2.83	2.86	2.96	3.08	2.97	2.87
MaxH	1.31	1.26	1.28	1.3	1.38	1.32	1.29

## Data Availability

The original contributions presented in this study are included in the article. Further inquiries can be directed to the corresponding author A.G.

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
