# Peer review of "Morphological, Morphometrical and Radiological Features of the Pelvic Limb Skeleton in African Green Monkeys (Chlorocebus sabaeus) from Saint Kitts and Nevis Islands"

_animals, 2025, doi:10.3390/ani15020209_

Round 1

Reviewer 1 Report

Comments and Suggestions for Authors

The authors describe the gross anatomy of the pelvic limb of a monkey species that has high similarity to human body, African green monkey (Clorocebus sabaeus). The work appears original, well written and scientifically sounds. The aim of the work was to study the skeleton of pelvic limbs of this species. The methods employed were the comparative anatomy after disection and the use of radiological projections. The images are of good quality and the numbering and legends useful.  The references appropiate. The description of the osteology is correct. I think that this work deserves to be published after a minor revision. In particular I suggest these minor points:

The authors should spend more information about the origin of the monkeys used and more information about the age and if they were male or female. 

The Figures should cited as:  (Figg. 2-4) instead of (Fig. 2; Fig. 3; Fig. 4).

Sometimes the word "condylus" is "condilus" (for example, line 303).

Author Response

The authors describe the gross anatomy of the pelvic limb of a monkey species that has high similarity to human body, African green monkey (Clorocebus sabaeus). The work appears original, well written and scientifically sounds. The aim of the work was to study the skeleton of pelvic limbs of this species. The methods employed were the comparative anatomy after disection and the use of radiological projections. The images are of good quality and the numbering and legends useful.  The references appropiate. The description of the osteology is correct. I think that this work deserves to be published after a minor revision. In particular I suggest these minor points:

Thank you!

The authors should spend more information about the origin of the monkeys used and more information about the age and if they were male or female. 

Notes on the sex and the origin of the skeletal pieces and the carcass was made

The Figures should cited as:  (Figg. 2-4) instead of (Fig. 2; Fig. 3; Fig. 4).

Sometimes the word "condylus" is "condilus" (for example, line 303).

made the corrections

Reviewer 2 Report

Comments and Suggestions for Authors

Dear authors,

I have been reading your manuscript on the Morphological, morphometrical and radiological features of the pelvic limb skeleton in African green monkeys with great interest. Below are my remarks and questions.

General editorial remark: The manuscript contains some typo’s, is not always consistent in the use of capital vs. lower case letters, and some double or triple spacings are present. These issues should be resolved during the final editing of the manuscript, should it be accepted.

Keywords

-       Vervet monkey is not mentioned a single time in the entire document (besides in the reference list)

Introduction

-       Line 40: “the primates” Which primates do you mean? Humans are also primates.

Materials and methods

-       Line 75: Do you know the sex of the specimens?

-       Line 76: Where did the animal from which the fresh cadaver was obtained come from and how did it die?

-       Line 93: Canon. Please provide the manufacturer, its city and country. Idem for Photoshop and the radiology device.

-       Line 86: Regarding the N.A.V., please have a fresh look at the terms provided in the manuscript. Many of these contain typo’s. 

-       General remark: Please provide some introductory sentences before popping up the Tables for the “Pelvis measurements” etc.

Results

-       General remark regarding the figure legends 1: Latin and English terminology is used intermingled in the figure legends. You have to choose one language. Regarding the English terms, please consult the human anatomical terminology TA2.

-       General remark regarding the figure legends 2: Carefully check the punctuation (spacing, comma, capital vs. lower case letters, etc.)

-       The figures have to be renumbered since figure numbering restarts after Figure 6.

-       Can the reader know which values are from male and which from female specimens?

-       Line 164: The obturator nerve innervates the adductor musculature, not the abductors.

-       Line 381: “digits digital area” Please rephrase because “digits” seems redundant.

-       Line 388: What do you mean with “only the proximal row bones are morphologically relevant”? Where lies its relevance? This should be discussed.

Discussion:

-       Line 486: haplorrhine

-       Lines 618-621: These sentences should be removed as they are part of the template.

Reference list

-       Please remove the double references such as Casteleyn et al. in number 47 and in number 58, and renumber the list accordingly.

Author Response

Dear authors,

I have been reading your manuscript on the Morphological, morphometrical and radiological features of the pelvic limb skeleton in African green monkeys with great interest. Below are my remarks and questions.

General editorial remark: The manuscript contains some typo’s, is not always consistent in the use of capital vs. lower case letters, and some double or triple spacings are present. These issues should be resolved during the final editing of the manuscript, should it be accepted.

Thank you for your attention and patience. We will double check again the text for typos and other errors

Keywords

-       Vervet monkey is not mentioned a single time in the entire document (besides in the reference list)

We have added or replaced the term vervet in the text, with the risk of not being all the times very clear using AGM, green monkey and vervet monkey to designate the same species

Introduction

-       Line 40: “the primates” Which primates do you mean? Humans are also primates.

Added the term non-human primates

Materials and methods

-       Line 75: Do you know the sex of the specimens?

Added the data in the Methodology section- 3 males and 2 female specimens

-       Line 76: Where did the animal from which the fresh cadaver was obtained come from and how did it die?

I also added some info about the cadaver donated by the Pathology Department. We had little info on this, as it has been preserved in the freezer for a long period as a roadkill. We noticed it was an integer adult male with no noticeable limb fractures.

-       Line 93: Canon. Please provide the manufacturer, its city and country. Idem for Photoshop and the radiology device.

Added the requested information

Although my personal knowledge in Latin is very reduced, the NAV nominations use  direcly the genitive forms for nouns (eg. os ilium as nominative but when referring to structures of ilium uses directly the genitive form- corp osis ilii, as -um becomes i). Saw the Human Terminology suggested by you where only nominatives are used and only at “others” appear this genitive nomination. If the reviewer is not agreeing to this, we can change the whole set of nominations.

-       General remark: Please provide some introductory sentences before popping up the Tables for the “Pelvis measurements” etc.

done

Results

-       General remark regarding the figure legends 1: Latin and English terminology is used intermingled in the figure legends. You have to choose one language. Regarding the English terms, please consult the human anatomical terminology TA2.

Same as in the lines above- most probably due to different referencing systems- NAV for veterinarians and Human Anatomy Terminology, there are slight differences. We think that the usage of NAV is more appropriate as we are pointing to quadrupedal locomotion species, much closer to our (veterinary) referencing system. We are totally aware of the confusion that it might create (eg. coxal tuberosity or sacral tuberosity in iliac wing, posterior superior and inferior spines vs ventral iliac spine; linea aspera and linea pectinea/medial+lateral linea aspera) for those used with human anatomical terms.

Generally we tried to harmonize the two sources, using mainly English nominations, removing, whenever possible, most of the Latin ones from figure legends

-       General remark regarding the figure legends 2: Carefully check the punctuation (spacing, comma, capital vs. lower case letters, etc.)

See above

-       The figures have to be renumbered since figure numbering restarts after Figure 6.

Most probably a technical glitch- alltogether we have 11 figures, numbered correctly in the original file

-       Can the reader know which values are from male and which from female specimens?

Added a note in the introductory part stating which code corresponds to female or male individuals

-       Line 164: The obturator nerve innervates the adductor musculature, not the abductors.

Thanks for your observation. Corrected

-       Line 381: “digits digital area” Please rephrase because “digits” seems redundant.

Corrected

-       Line 388: What do you mean with “only the proximal row bones are morphologically relevant”? Where lies its relevance? This should be discussed.

 We focused on these two bones as we considered these easy to identify and recognize even as disparate bone pieces. Some other tarsals were missing in some of the investigated individuals. A note was inserted to replace the earlier starting sentence.

Discussion:

-       Line 486: haplorrhine

-       Lines 618-621: These sentences should be removed as they are part of the template.

 Corrections made

Reference list

-       Please remove the double references such as Casteleyn et al. in number 47 and in number 58, and renumber the list accordingly.

Correction made

Reviewer 3 Report

Comments and Suggestions for Authors

Dear authors,

The manuscript entitled “Morphological, morphometrical and radiological features of the pelvic limb skeleton in African green monkeys (Chlorocebus sabaeus) from Saint Kitts and Nevis islands” is a great contribution to the anatomical knowledge in non-human primates. Your anatomical descriptions are excellent, supported by very good photographs and radiographs of the pelvic limb bones. The detailed recognition of the bony reliefs is correct and well-indicated. The discussion of the comparative aspects with other non-human primates is well performed. However, I suggest performing major and minor revisions to improve your manuscript.

Major changes

I recommend changing the words “morphology” to anatomy, “morphological” to anatomical, etc, because morphology involves histology and embryology. Therefore, based on the content of your manuscript, it is only about gross anatomy.

It is important to know the sex of the specimens to review their differences, mainly at the pelvis level.

The name of the measurements should be adapted to the anatomical terms to quadrupeds (NAV, 2017) since the terms superior, inferior, anterior, and posterior are only used in humans. It will improve the application of your anatomical description in morphometry. Several articles on non-human primates use the anatomical terms to humans, so you can include the measurement names between parentheses. You should include the descriptive statistics of the bone measurements and after compare them with a statistic test. For example, compare the length of the bones to review if there are significant differences among specimens or long bones (E.g. tibia and femur).

One of the highlights of this manuscript is the report of the sesamoid bones different from patella. Please include a discussion comparing the sesamoid bones with other primates or other species, such as the presence of the sesamoid bones of the gastrocnemius muscle and the sesamoid bones of the foot. In Lemur catta, the sesamoid bone of the popliteus muscle is present in the radiographs, although the authors did not indicate it (Makungu et al., 2013). In several manuscripts, the sesamoid bones of the gastrocnemius muscle are observed in the radiographs, but the authors did not describe them (Souza Siragusi et al., 2020; de La Salles et al., 2023; Makungu et al., 2013). There are two anatomical books about two Neotropical primates where the appendicular bones are described with the support of radiographs (Montilla Rodríguez et al., 2023; Vélez García et al., 2021). You could review them since both are also two quadrupedal species. Even the sesamoid bones of the gastrocnemius and popliteus muscles, and the proximal sesamoid bones at the level of the metatarsophalangeal joints are described in white-footed tamarin (Saguinus leucopus).

Review the following references:

1.     de Souza Siragusi, R. H., Rahal, S. C., da Silva, J. P., Mamprim, M. J., Rolim, L. S., Teixeira, C. R., ... & Monteiro, F. O. (2020). Radiographic evaluation of the forelimbs and hind limbs of marmosets (Callithrix spp.). Journal of Medical Primatology49(2), 71-78.

2.     Stevens, J. L., Mitton, S., & Edgerton, V. R. (1972). Gross anatomy of hindlimb skeletal muscles of the Galago senegalensis. Primates13, 83-101

3.     Hepburn, D. (1892). Comparative anatomy of the muscles and nerves of the superior and inferior extremities of the anthropoid apes: Part II. Journal of Anatomy and Physiology26(Pt 3), 324.

4.     Casteleyn, C., Bosmans, M., Muylle, S., & Bakker, J. (2024). The Foot Musculature of the Rhesus Monkey (Macaca mulatta): An Anatomical Study. Anatomia3(4), 256-276.

5.     Casteleyn, C., Robin, N., & Bakker, J. (2023). Topographical Anatomy of the Rhesus Monkey (Macaca mulatta)—Part II: Pelvic Limb. Veterinary sciences10(3), 172.

6.     Montilla Rodríguez, M. A.; Blanco Rodríguez, J.C. & Sánchez Rojas, P.B. (2023). Atlas osteológico de Plecturocebus caquetensis. Editorial Universidad de la Amazonia.

7.     Vélez-García, J. F., Castañeda-Herrera, F., Ospina-Herrera, O., Villamil-González, D. C., & Monroy-Cendales, M. J. (2021). Atlas anatómico del tití gris (Saguinus leucopus). Manizales: ISAGEN-CORPOCALDAS-UNIVERSIDAD DEL TOLIMA.

Minor changes

Change hindlimb to pelvic limb along the manuscript

Lines

2: Italizice Chlorocebus sabaeus

14: Imagistic or Imagenological features?

17: change morphological to anatomical

21 include the scientific name after the common name. Eg.: African green monkey (Chlorocebus sabaeus).

29: change inferior to distal

31: change greater tubercle to greater trochanter

35: include keywords that are not present in the title

40: The humans are primates. …between non-human primates and humans…

41: idem: non-human primates and humans

44: front limbs or thoracic limbs?

49: include the scientific name after the common name because it is the first time that you name it.

54: after the first use of the complete scientific name, you should abbreviate the name C. sabaeaus. Thus, abbreviate from this line if you include the anterior comment.

79: How many males and females?

115: Use only the acronym AGM because you named it in the introduction or use the abbreviated form.

157: …dorsal to the acetabulum…?

216, 277, 353: acronym or abbreviated form

226: what indicates the white circle with the dashed line of the image D?

255: Did you find the sesamoid bone of the popliteal muscle “Os sesamoideum m. poplitei”? If not, please include a sentence reporting the absence.

257, 274: change the term fabella to sesamoid bones of the gastrocnemius muscle.

277: Is it rectangular? I see in the figure 7 an oval form.

285: Figure 7. A. Cranial surface

315: Figure 8

315: include the views of the fibula: lateral and medial?

329: change tibial crest to cranial margin. Tibial crest is not at the NAV.

369: change hind paw skeleton to pes skeleton

443: I can see the proximal sesamoid bones in all metatarsophalangeal joints.

464-467: Please specify if there are proximal sesamoid bones on the plantar side of all metatarsophalangeal joints. Is the distal sesamoid bone in all digits or only in the digi I?

470: Use the most current phylogenetic classification of primates. I think that the most current is the study of Shao et al. (2023)

Shao, Y., Zhou, L., Li, F., Zhao, L., Zhang, B. L., Shao, F., ... & Wu, D. D. (2023). Phylogenomic analyses provide insights into primate evolution. Science380(6648), 913-924.

489: abbreviated form C. sabaeaus; cranio-caudal direction?

530: this reference is only about lemur.

612: Include the last name of the first author Takehisa [91]

618-621: I do not understand Why you are including this paragraph?

Author Response

Dear authors,

The manuscript entitled “Morphological, morphometrical and radiological features of the pelvic limb skeleton in African green monkeys (Chlorocebus sabaeus) from Saint Kitts and Nevis islands” is a great contribution to the anatomical knowledge in non-human primates. Your anatomical descriptions are excellent, supported by very good photographs and radiographs of the pelvic limb bones. The detailed recognition of the bony reliefs is correct and well-indicated. The discussion of the comparative aspects with other non-human primates is well performed. However, I suggest performing major and minor revisions to improve your manuscript.

Thank you for your kindness!

Major changes

I recommend changing the words “morphology” to anatomy, “morphological” to anatomical, etc, because morphology involves histology and embryology. Therefore, based on the content of your manuscript, it is only about gross anatomy.

Done the replacement of the term morphology/morphological with anatomy/anatomical

It is important to know the sex of the specimens to review their differences, mainly at the pelvis level.

We added the data on sex of the individuals in the introductory part. The different state of preservation, not only for pelvis but for other segments, made some of the measurements impossible (for instance there was an old arthritic individual showing a significant amount of exostotic lesions). This, and the very small sample size, made the attempt on separating the sex-related features quite useless, preferring not to continue further in this initiative for the moment.

The name of the measurements should be adapted to the anatomical terms to quadrupeds (NAV, 2017) since the terms superior, inferior, anterior, and posterior are only used in humans. It will improve the application of your anatomical description in morphometry. Several articles on non-human primates use the anatomical terms to humans, so you can include the measurement names between parentheses. You should include the descriptive statistics of the bone measurements and after compare them with a statistic test. For example, compare the length of the bones to review if there are significant differences among specimens or long bones (E.g. tibia and femur).

We tried to apply the NAV terms in most cases (as this seems to be the best choice for these quadripedal animals), but in some cases (as you also mention) some human-anatomy terms remained as they are used more frequently in scientific literature. More than that, the other reviewer insisted on the usage of human AT (most probably a human doctor or biologist) and we also made a plea for the usage of NAV instead of the suggested nominations. Hope that these opposing ideas will not influence the decisions in regards to this paper. The same stands for the standardized (?) list of measurements used in human anatomy that are  most of the times used also for primates. As you probably know, the standard osteometric measurements for animal anatomy is less comprehensive (that was in fact our starting point) and led us to this other set of more detailed measurements. For the sake of using the same reference points in terms of nominations, we left the codes unchanged.

One of the highlights of this manuscript is the report of the sesamoid bones different from patella. Please include a discussion comparing the sesamoid bones with other primates or other species, such as the presence of the sesamoid bones of the gastrocnemius muscle and the sesamoid bones of the foot. In Lemur catta, the sesamoid bone of the popliteus muscle is present in the radiographs, although the authors did not indicate it (Makungu et al., 2013). In several manuscripts, the sesamoid bones of the gastrocnemius muscle are observed in the radiographs, but the authors did not describe them (Souza Siragusi et al., 2020; de La Salles et al., 2023; Makungu et al., 2013). There are two anatomical books about two Neotropical primates where the appendicular bones are described with the support of radiographs (Montilla Rodríguez et al., 2023; Vélez García et al., 2021). You could review them since both are also two quadrupedal species. Even the sesamoid bones of the gastrocnemius and popliteus muscles, and the proximal sesamoid bones at the level of the metatarsophalangeal joints are described in white-footed tamarin (Saguinus leucopus).

As our study relies on skeletal specimens from a collection, we examined specimens that were prepared by someone else. We did have info on sex and age, but sometimes the collections were not complete- missing bones, fragmented specimens etc. The only carcass we examined was assessed radiologically and highlighted clearly the existence of the femoral sesamoids (gastrocnemian). We did not perform a further investigation to check for cyamella (even as we knew about their possible presence). We inserted a paragraph mentioning the sesamoids in the last part of our paper. The suggested references are very useful. Thank you!

Review the following references:

  1. de Souza Siragusi, R. H., Rahal, S. C., da Silva, J. P., Mamprim, M. J., Rolim, L. S., Teixeira, C. R., ... & Monteiro, F. O. (2020). Radiographic evaluation of the forelimbs and hind limbs of marmosets (Callithrix spp.). Journal of Medical Primatology, 49(2), 71-78.
  2. Stevens, J. L., Mitton, S., & Edgerton, V. R. (1972). Gross anatomy of hindlimb skeletal muscles of the Galago senegalensis. Primates, 13, 83-101
  3. Hepburn, D. (1892). Comparative anatomy of the muscles and nerves of the superior and inferior extremities of the anthropoid apes: Part II. Journal of Anatomy and Physiology, 26(Pt 3), 324.
  4. Casteleyn, C., Bosmans, M., Muylle, S., & Bakker, J. (2024). The Foot Musculature of the Rhesus Monkey (Macaca mulatta): An Anatomical Study. Anatomia, 3(4), 256-276.
  5. Casteleyn, C., Robin, N., & Bakker, J. (2023). Topographical Anatomy of the Rhesus Monkey (Macaca mulatta)—Part II: Pelvic Limb. Veterinary sciences, 10(3), 172.
  6. Montilla Rodríguez, M. A.; Blanco Rodríguez, J.C. & Sánchez Rojas, P.B. (2023). Atlas osteológico de Plecturocebus caquetensis. Editorial Universidad de la Amazonia.
  7. Vélez-García, J. F., Castañeda-Herrera, F., Ospina-Herrera, O., Villamil-González, D. C., & Monroy-Cendales, M. J. (2021). Atlas anatómico del tití gris (Saguinus leucopus). Manizales: ISAGEN-CORPOCALDAS-UNIVERSIDAD DEL TOLIMA.

Minor changes

Change hindlimb to pelvic limb along the manuscript

 Change made

Lines

2: Italizice Chlorocebus sabaeus

done

14: Imagistic or Imagenological features?

We left it as imagistic

17: change morphological to anatomical

Made the change in most of the places throughout text

21 include the scientific name after the common name. Eg.: African green monkey (Chlorocebus sabaeus).

done

29: change inferior to distal

done

31: change greater tubercle to greater trochanter

done

35: include keywords that are not present in the title

done

40: The humans are primates. …between non-human primates and humans…

corrected

41: idem: non-human primates and humans

corrected

44: front limbs or thoracic limbs?

Replaced the term with thoracic limb

49: include the scientific name after the common name because it is the first time that you name it.

done

54: after the first use of the complete scientific name, you should abbreviate the name C. sabaeaus. Thus, abbreviate from this line if you include the anterior comment.

done

79: How many males and females?

Inserted a short paragraph about the sex of the animals in the Materials section

115: Use only the acronym AGM because you named it in the introduction or use the abbreviated form.

157: …dorsal to the acetabulum…?

rephrased

216, 277, 353: acronym or abbreviated form

226: what indicates the white circle with the dashed line of the image D?

Added in figure legend- the extensor’s fossa

255: Did you find the sesamoid bone of the popliteal muscle “Os sesamoideum m. poplitei”? If not, please include a sentence reporting the absence.

Did that in a short explanatory paragraph in the discussion section. We did not identify the popliteal sesamoid bone on the radiograph

257, 274: change the term fabella to sesamoid bones of the gastrocnemius muscle.

As the term fabella is very widely used in veterinary medicine, we kept this nomination throughout the text but added the suggested term every time the nomination is used using brackets- fabella(sesamoid bones of gastrocnemius)

277: Is it rectangular? I see in the figure 7 an oval form

We changed the term to describe the complex shape of the patella. We have used the term pentagonal-shaped as oval is closer to the description in patella for carnivores. Here we consider the existence of some lateral margins and the dorsal margin as the base of the irregular pentagon.

285: Figure 7. A. Cranial surface

corrected

315: Figure 8

315: include the views of the fibula: lateral and medial?

Added a note in legend

329: change tibial crest to cranial margin. Tibial crest is not at the NAV.

corrected

369: change hind paw skeleton to pes skeleton

corrected

443: I can see the proximal sesamoid bones in all metatarsophalangeal joints.

Changed the phrasing in order to clarify the fact that all joints show the presence of the sesamoid bones

464-467: Please specify if there are proximal sesamoid bones on the plantar side of all metatarsophalangeal joints. Is the distal sesamoid bone in all digits or only in the digi I?

Changed the text so we clarified the fact that all joints have the sesamoids- both proximal and distal.

470: Use the most current phylogenetic classification of primates. I think that the most current is the study of Shao et al. (2023)

Shao, Y., Zhou, L., Li, F., Zhao, L., Zhang, B. L., Shao, F., ... & Wu, D. D. (2023). Phylogenomic analyses provide insights into primate evolution. Science, 380(6648), 913-924.

489: abbreviated form C. sabaeaus; cranio-caudal direction?

Made some correction of terms

530: this reference is only about lemur.

Double checked

612: Include the last name of the first author Takehisa [91]

Made the change in accordance to the MDPI requirements

618-621: I do not understand Why you are including this paragraph?

Typo…forgot to erase from the template provided by the journal

Round 2

Reviewer 3 Report

Comments and Suggestions for Authors

Dear authors, the new version of your manuscript only has minor changes. I recommended in my first review major changes that were not performed. I think important to include the descriptive statistics of your morphometric data. The statistic study would improve your comparative descriptions to explain if there are or are no significant differences among bones, specimens, etc. This new version has anatomical terms that should not be used in the limbs of quadrupedal animals (e.g. anterior and posterior). The anatomical terms to the human are different to the limbs. Several terms were well written in the first version of your manuscript. The term ischiatic was well written. Please review a current anatomical veterinary textbook (review the Evans and Miller's anatomy of the dog) and use the correct terminology based on the NAV 2017. The keywords has words of the title. The extensor fossa is only in animals where the origin tendon of the m. Extensor digitorum longus originates from here. My experience dissecting non-human primates, the extensor digitorum longus muscle originates from the Tibia. Please review again if actually there is a extensor fossa. The terms fabella and cyamella should not used, these are not present at the NAV. Continue the norms of the NAV. The metatarsal sesamoid bones are not at the NAV. The correct terms are proximal sesamoid bones. Are you sure that there are distal sesamoid bones in all digits? I cannot see these bones at the radiography of the pes. Please point on the proximal sesamoid bones in the radiography of the pes.

Line 37: the terms proximal and distal metatarsian bones or proximal sesamoid bones?

Line 388: hind pes or pes skeleton?

Author Response

Dear Reviewer,

Thank you very much for your second set of notes. As you might have read in our initial comment, we tried to cope with comments leading 2 divergent directions.

Hope these corrections will add, as you said, some new and satisfactory additions to our initial text.

Dear authors, the new version of your manuscript only has minor changes. I recommended in my first review major changes that were not performed. I think important to include the descriptive statistics of your morphometric data. The statistic study would improve your comparative descriptions to explain if there are or are no significant differences among bones, specimens, etc.

We initially avoided stats as the number of measurements available was very low. As you suggested, we added some average data, in our attempt to highlight (if possible) some metrical differences that might be revealed from this very reduced set of data. We are aware that a n=5 (maybe +/- 2-3) or a little more is not satisfactory for a sound statistical interpretation, but some of the figures may indicate something, as you can see in the newly inserted data. What we saw is most probably the difference related to sexual dimorphism (-up to 20% for females) or some facts that might not be related to this fact.

This new version has anatomical terms that should not be used in the limbs of quadrupedal animals (e.g. anterior and posterior). The anatomical terms to the human are different to the limbs. Several terms were well written in the first version of your manuscript. The term ischiatic was well written. Please review a current anatomical veterinary textbook (review the Evans and Miller's anatomy of the dog) and use the correct terminology based on the NAV 2017. We reapproached the text and modified the terms (some modified due to demands of the second reviewer that insisted on using the human AT)

The keywords has words of the title.

Removed from keywords

 The extensor fossa is only in animals where the origin tendon of the m. Extensor digitorum longus originates from here. My experience dissecting non-human primates, the extensor digitorum longus muscle originates from the Tibia. Please review again if actually there is a extensor fossa. The terms fabella and cyamella should not used, these are not present at the NAV. Continue the norms of the NAV.

We revised the interpretation and noted on the image and in the text the structure that we consider to be a ligamentary fossa (as illustrated in Barone)

The metatarsal sesamoid bones are not at the NAV. The correct terms are proximal sesamoid bones. Are you sure that there are distal sesamoid bones in all digits? I cannot see these bones at the radiography of the pes. Please point on the proximal sesamoid bones in the radiography of the pes.

Line 37: the terms proximal and distal metatarsian bones or proximal sesamoid bones?

Clarified the sesamoid bone issue. Proximal and distal terms were used. Noted the presence of the proximal ones in all digits and the identification of distal ones in case of the 1-st digit only. Addition of references and similarities, as suggested in your first comment, were added (Souza 2020 and Makungu articles). 

Round 3

Reviewer 3 Report

Comments and Suggestions for Authors

The article has been improved. I only have one major correction and several minor corrections that should be made in the article.

Major correction:

Please include in Material and Methods, the statistical method to report significant difference in the line 231 or no in the line 486.

Minor corrections:

Lines:

39: change the term "pelvic limb" to "radiography"

36, 250, 325, 611, change "gastrocnemian muscle" to "gastrocnemius muscle"

36, 37: change the term "sesamoidian" to "sesamoid"

50, 51, 88, 125: change hind limb to pelvic limb

22, 23, 100, 549, 686: Include the term "limb" after "pelvic" (pelvic limb)

104: planto-dorsal

117-121: F15, F16, T13: anteroposterior to cranio-caudal

F21: posterior condyle or caudal margin of the lateral or medial condyle? anterior margin or cranial margin of the trochlea?

122: change ...the most superior... to ...the most proximal...

191: change anterio to cranio

250, 251: change bones to bone. (...sesamoid bone)

326: change dorsal to proximal

355: change facet to surface

680: reference 92 is not about the white-footed tamarin. The correct reference, I included in the first review (reference 7: Vélez-García et al. 2021). However, conserve reference 92 to include other Callitrichids.

Author Response

Dear Reviewer,

Thank you again for your attention and patience, Your contribution was important for the quality of our paper.

As far as your major notes are concerned, we completed the Materials section with one phrase mentioning the basic Google Sheets stats we performed. It was a simple average and sd calculation we used for that reduced set we had. 

The minor corrections were performed, as suggested

The Bibliography was completed with Velez-Garcia reference, inserted along the previous one, so it was not a major intervention to do.

Thanks again!

Regards,

Round 4

Reviewer 3 Report

Comments and Suggestions for Authors

Most suggestions were performed. The reference 92 is wrong because is about a research in another species (Tamandua) of Vélez-García et al. The correct reference is about the white footed tamarin that is a book in Spanish: Vélez- García JF, Castañeda- Herrera F, Ospina- Herrera O, Villamil- González DC, Monroy- Cendales MJ. Atlas Anatómico Del Tití Gris (Saguinus Leucopus). Artes Pixelar SAS; 2021. 

After perform this minor correction the article is ready to publish.

Congratulations